

# Carbon dioxide and methane exchange of a patterned subarctic fen during two contrasting growing seasons

Lauri Heiskanen[1], Juha-Pekka Tuovinen[1], Aleksi Räsänen[2], Tarmo Virtanen[2], Sari Juutinen[2], Annalea Lohila[1], Timo Penttilä[3], Maiju Linkosalmi[1], Juha Mikola[2], Tuomas Laurila[1], Mika Aurela[1]

[1]Finnish Meteorological Institute, Helsinki, Finland
[2]Ecosystems and Environment Research Programme, Faculty of Biological and Environmental Sciences, University of Helsinki, Finland
[3]Natural Resources Institute Finland (LUKE), Helsinki, Finland

*Correspondence to*: Lauri Heiskanen (lauri.heiskanen@fmi.fi)

**Abstract.**

The patterned microtopography of subarctic mires generates a variety of environmental conditions, and carbon dioxide ($CO_2$) and methane ($CH_4$) dynamics vary spatially among different plant community types. We studied the $CO_2$ and $CH_4$ exchange between a subarctic fen and the atmosphere at Kaamanen in northern Finland based on flux chamber and eddy covariance measurements in 2017–2018. We observed strong spatial variation in carbon dynamics between the four main plant community types (PCTs) studied, which were largely controlled by water table level and differences in vegetation composition. The ecosystem respiration (ER) and gross primary productivity (GPP) increased gradually from the wettest PCT to the drier ones, and both ER and GPP were larger for all PCTs during the warmer and drier growing season 2018. We estimated that in 2017 the growing season $CO_2$ balances of the PCTs ranged from -20 g C m$^{-2}$ (*Trichophorum* tussock PCT) to 64 g C m$^{-2}$ (string margin PCT), while in 2018 all PCTs were small $CO_2$ sources (10–22 g C m$^{-2}$). We observed small growing season $CH_4$ emission sums (< 1 g C m$^{-2}$) from the driest PCT, while the other three PCTs had significantly larger emissions sums (mean 7.9, range 5.6–10.1 g C m$^{-2}$) during the two growing seasons. Compared to the annual $CO_2$ balance (-8.5 ± 4.0 g C m$^{-2}$) of the fen in 2017, in 2018 the annual balance (-5.6 ± 3.7 g C m$^{-2}$) was affected by an earlier onset of photosynthesis in spring, which increased the $CO_2$ sink, and a drought event during summer, which decreased the sink. The $CH_4$ emissions were also affected by the drought. The annual $CH_4$ balance of the fen was 7.3 ± 0.2 g C m$^{-2}$ in 2017 and 6.2 ± 0.1 g C m$^{-2}$ in 2018. Thus, the carbon balance of the fen was close to zero in both years. The PCTs adapted to drier conditions provided resilience to carbon loss due to water level draw down.

## 1 Introduction

Northern mires have sequestered substantial amounts of atmospheric carbon (C) since the last glacial period. The C storage of these peat soils has been estimated to be 415 ± 150 Pg of C (Hugelius et al., 2020), which adds up to about 30% of the global soil C. This C storage has accumulated through the photosynthetic fixation of carbon dioxide ($CO_2$) by mire vegetation, which in the long term has been larger than the release of C through plant respiration and peat decomposition. In the short term, however, the C balance of a mire can switch from a sink to a source, as the annual C accumulation rate is sensitive to variations in moisture conditions and temperature (Alm et al., 1999; Bubier et al.,



2003; Lindroth et al., 2007; Olefeldt et al., 2017) and to the length of the snow-free period (Aurela et al., 2004; Lund et al., 2012). Understanding the annual variability in peatland C dynamics is essential, as the subarctic and arctic regions warm rapidly, two to three times as fast as the rest of the world (IPCC 1.5 °C Special report, 2018). This is projected to result in increased evapotranspiration and altered precipitation patterns, affecting in turn the C balance of

mires (Tarnocai, 2006).

Subarctic mires endure long winters and relatively short growing seasons and have near-zero mean annual air temperatures. The net response of ecosystem C exchange to annual weather conditions depends on multiple processes and is thus hard to predict. In particular, the timing of soil thaw and snow melt has been shown to impact the growing season length and consequently the annual net $CO_2$ uptake of mires, because earlier springs advance the timing of bud

burst (Aurela et al., 2004; Grøndahl et al., 2008). In addition, warm springs increase microbial activity, ecosystem respiration (Lafleur et al., 2005) and, if anoxia prevails, methanogenesis and methane ($CH_4$) emissions to the atmosphere (Kim et al., 1999).

Ecosystem respiration, gross primary production and net $CH_4$ production in mires depend strongly on water table level (WTL), which determines the depth of oxic and anoxic layers in the peat. Microforms with varying WTL, e.g.

hummocks and hollows, create different habitats for plant and microbe communities (Wieder et al., 2006). Peat decomposition is greater in the oxic layer above the WTL (Silvola et al., 1996), and deeper oxic layers also enhance plant production as roots require aeration (Bubier et al., 2003). Consequently, the net $CO_2$ exchange in northern mires not only varies among different sites (Bubier et al., 1998; Frolking et al., 1998; Lindroth et al., 2007) and from year to year (Aurela et al., 2004; Lund et al., 2012) but also among the microforms and plant communities within a site

(Bubier et al. 1998; Alm et al. 1999; Heikkinen et al., 2004). However, there does not seem to be a clear universal spatial pattern in relation to changing moisture conditions (Alm et al., 1999; Waddington and Roulet, 2000; Laine et al., 2007a; Strack and Waddington, 2007; Maanavilja et al. 2011; Korrensalo et al., 2019). In contrast, $CH_4$ emissions are generally larger from wet than dry plant communities, as they depend on the balance between microbial production in anoxic conditions and oxidation above the WTL (Saarnio et al., 1997; Segers, 1998; Alm et al., 1999; Heikkinen et

al., 2004; Wieder et al., 2006; Laine et al., 2007b), but there is additionally marked small-scale spatial variation related to nutrient and substrate availability and plant species composition (Svensson and Rosswall, 1984; Kettunen, 2003; Christensen et al., 2004; Dorodnikov et al., 2011).

High temperatures with water level drawdown have been reported to decrease $CO_2$ uptake (Chivers et al., 2009; Munir et al., 2014) and $CH_4$ emissions (Peltoniemi et al., 2016; Olefeldt et al., 2017) of mires. During the summer of 2018,

a large-scale heatwave and drought took place in north-western Europe, including northern Finland (Lehtonen and Pirinen, 2019a,b). Rinne et al. (2020) found that this drought reduced $CO_2$ uptake and $CH_4$ emissions, as compared to a reference year, on most of the Fennoscandian mires studied. However, the magnitude of this effect varied among the mires, as did the duration and severity of the drought.

In this study, we examine the $CO_2$ and $CH_4$ exchange between the subarctic Kaamanen fen and the atmosphere during

two contrasting years (2017 and 2018). The site was included in the synthesis of Rinne et al. (2020) that applied spatially averaged eddy covariance (EC) data. However, the annual variation in the plant-community-scale C exchange has not been investigated previously, and it is unknown how different communities react to water level drawdown



during drought events. Thus, our specific objective is to study the small-scale $CO_2$ and $CH_4$ flux variation in response to moisture conditions and compare this to the ecosystem-scale response. We (1) utilise the EC flux measurement technique to detect the ecosystem-scale variation in C exchange, (2) study how C exchange varies spatially and temporally among different plant communities by using manual flux chamber measurements and (3) determine the main environmental factors controlling the C fluxes by means of a linear mixed effects model. We will also put our results in the context of the earlier gas exchange data from this measurement site.

## 2 Materials and methods

### 2.1 Study site

The study took place at a patterned mesotrophic fen at Kaamanen in northern Finland (69° 8.435' N, 27° 16.189' E, 155 m a.s.l.). The average annual mean temperature at the Inari Ivalo weather station, 59 km south of Kaamanen, during the 30-yr reference period of 1981–2010 was -0.4 °C, and the corresponding mean annual precipitation sum was 472 mm (Pirinen et al., 2012). A major part of the fen is patterned with strings and flarks (or hummocks and hollows, respectively). A few metre-wide hummocks surround wet flarks and form strings, up to a few tens of metres in length that sprawl through the fen. The strings are elevated from the water table by 0.3–0.8 m and contain ice lenses that can remain frozen until late summer (Aurela et al., 2001). The site has no permafrost, even though it locates 300 km north of the Arctic Circle and the nearest isolated permafrost palsas are currently about 50 km north of the site. The peat depth within the study area is 1–2 m (Piilo et al., in press).

The vegetation within the patterned fen, which is the focus of our measurements, can be divided into four main plant community types (PCTs): (1) *Ericales-Pleurozium* on the dry string tops (ST), (2) *Betula nana-Sphagnum* on the string margins (SM), and (3) *Trichophorum* tussock (TT) and (4) flark (F) communities in the wet hollows (Fig. 1, Table 1) (Maanavilja et al., 2011). The areal coverage of different land cover types at the site has been estimated with high spatial resolution remote sensing by Räsänen and Virtanen (2019) (Table 1). A few sporadic *Betula pubescens* and *Pinus sylvestris* trees are growing on the driest parts of the fen.

A stream flows through the fen from north to south, but there is also nearly continuous surface water flow across the peatland along the elevation gradient. The fen is flooded during the spring thaw, but the magnitude of flooding varies annually. The string tops, however, are separated from the water table of rest of the fen (Fig. 1) and therefore mainly receive nutrients from precipitation instead of the lateral inflow.

The C exchange of the fen has been studied extensively during the past few decades (Aurela et al., 1998, 2001, 2002, 2004; Hargreaves et al., 2001; Laurila et al., 2001; Heikkinen et al., 2002; Maanavilja et al., 2011; Kross et al., 2016; Rinne et al., 2020; Piilo et al., in press). The ecosystem-atmosphere $CO_2$ exchange has been measured with the EC technique continuously since 1997 and $CH_4$ exchange since 2010. On average, the fen has been estimated to be a small (approximately 15 g C m$^{-2}$ yr$^{-1}$) atmospheric C sink during the measurement periods 1997–2002 and 2011–2016 (Aurela et al., 2004, and unpublished data), but the C exchange varies among the microforms: the driest plant communities act as the weakest net $CO_2$ sink (Heikkinen et al., 2002, Maanavilja et al., 2011) and $CH_4$ source (Heikkinen et al., 2002).





**Table 1:** Vegetation composition of the four main fen plant communities and their areal coverage inside a 200 m radius around the
eddy covariance tower.

| Plant community type | Dominant species | Coverage [%] |
|---|---|---|
| String top (ST) | *Vaccinium spp., Empetrum nigrum,* Lichens, *Pleurozium schreberi, Rubus chamaemorus* | 15 |
| String margin (SM) | *Carex spp., Eriophorum vaginatum, Sphagnum warnstorfii, Betula nana* | 13 |
| *Trichophorum* tussock (TT) | *Trichophorum cespitosum, Campylium stellatum* | 9 |
| Flark (F) | *Scorpidium scorpioides, Carex limosa* | 34 |

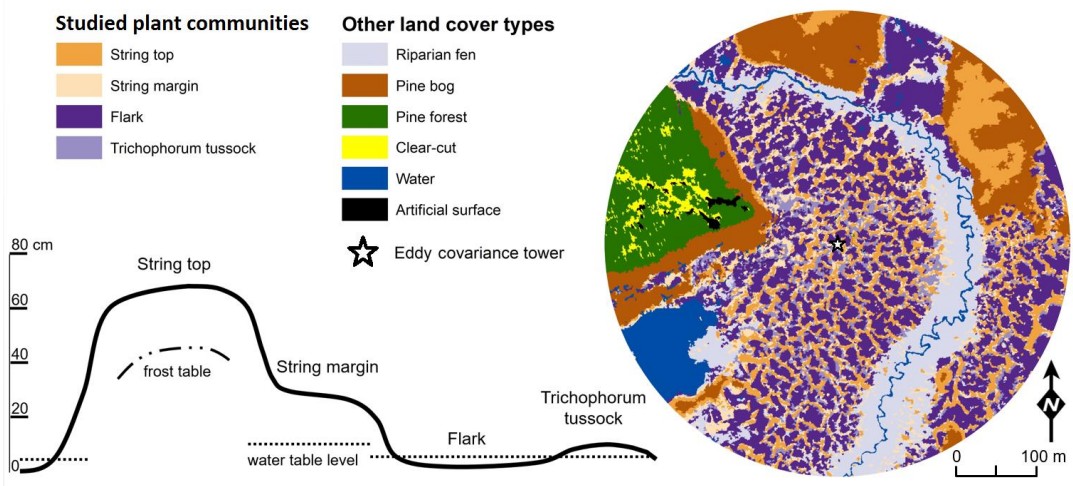

**Figure 1:** The four dominant plant communities (Table 1) within the main eddy covariance source area and the other land cover
types at the Kaamanen site. The land cover map is centred at the EC tower. The schematic cross-section of the fen microtopography
shows the average position of the studied plant community types (based on Maanavilja et al. (2011)).

## 2.2 Flux measurements

### 2.2.1 Chamber measurements

A total of 17 chamber flux measurement plots were chosen to represent the four PCTs, five plots for the SM
communities and four plots each for the ST, TT and F communities. Eight of the 17 aluminium collars (60 cm x 60
cm) were installed during the first days of June 2017, during the soil thawing, to accompany the collars that were
already installed during previous years. The collars were positioned within 50 m from the instrument booth (marked
with a star in Fig. 1).

The $CO_2$ and $CH_4$ fluxes between the ecosystem and atmosphere were measured with manual flux chambers biweekly
during the growing seasons of 2017 and 2018: six times between 12 June and 11 October 2017, and seven times



between 31 May and 4 September 2018. All chamber measurements were conducted between 9 a.m. and 4 p.m. local winter time.

The measurements were conducted with a transparent polycarbonate chamber (width x depth x height = 60 cm x 60 cm x 30 cm in 2017 and 60 cm x 60 cm x 40 cm in 2018). The chamber was connected with a 50 m long inlet tube (Teflon, inside diameter 3.1 mm) to a closed-path infrared gas analyser (Picarro G2401, Picarro Inc, USA) to detect the changes in $CO_2$, $CH_4$ and $H_2O$ mixing ratios in its airspace. The chamber air was mixed with a battery-driven fan. The chamber closure time for each measurement was 2 min. Air temperature inside the chamber and soil temperature at the 10 cm depth was measured at each plot on the chamber flux measurement days with IKES Pt100 sensors. Soil moisture was measured at the plots with a ML3 ThetaProbe sensor (Delta-T Devices Ltd, England) simultaneously with the chamber flux measurements. Photosynthetic photon flux density (PPFD) was measured during the chamber closures with PQS1 PAR Quantum sensor (Kipp & Zonen, Delft, The Netherlands) on top of the chamber.

The flux was determined by measuring the $CO_2$ and $CH_4$ mixing ratio change within the chamber, first in ambient light, then in one or two reduced light conditions and lastly in complete darkness; the amount of incoming solar radiation was reduced by 40–50%, 75–90% and 100%, respectively. The chamber was lifted off the collar between the measurements to restore the ambient gas concentration inside the chamber. Due to a small lag until the data can be accepted for $CO_2$ and $CH_4$ flux calculation only the data from the final 1.5 min of the 2 min closure time was used. The $CO_2$ and $CH_4$ fluxes were calculated as

$$F = \frac{p \times M \times V}{R \times T \times A} \times \frac{dc}{dt} \qquad (1)$$

where $p$ is atmospheric pressure, M is the molecular mass of $CO_2$ (44.01 g mol$^{-1}$) or $CH_4$ (16.04 g mol$^{-1}$), R is the universal gas constant (8.314 J mol$^{-1}$ K$^{-1}$), $T$ is the mean air temperature during chamber closure, V is the chamber volume, A is the chamber base area, and $\frac{dc}{dt}$ is the mean $CO_2$ or $CH_4$ mixing ratio change in time calculated with linear regression based on ordinary least squares. The mixing ratio is expressed with respect to dry air, so no correction for water vapour dilution was necessary. A micrometeorological sign convention was used: a positive flux indicates a flux from the ecosystem to the atmosphere (emission), and a negative flux indicates a flux from the atmosphere into the ecosystem (uptake).

For estimating the PCT-specific $CH_4$ flux time series from the chamber measurements, a mean flux was calculated for each of the 17 chamber plots from the two to four chamber closures that were conducted during each measurement day. The measurement data were screened to ensure that they were unaffected by $CH_4$ ebullition events and disturbances induced by a closing chamber. The criteria for discarding measurements were: $CH_4$ mixing ratio was > 5 ppm at the start of the closure, or the normalised root mean square error of the linear regression fit was > 0.02, or there was an obvious nonlinearity in the time series. The number of rejected/total data for each PCT were 27/148 (F), 4/150 (TT), 41/188 (SM) and 4/147 (ST).

### 2.2.2 Eddy covariance measurements

The eddy covariance (EC) measurements were conducted on a tower 5 m above the mean fen surface. The EC system consisted of a three-dimensional anemometer (USA-1, METEK Meteorologische Messstechnik GmbH, Germany), a closed-path infrared gas analyser for $CO_2$ and $H_2O$ mixing ratios (LI-7000, LI-COR Biosciences, USA) and a laser-





based gas analyser for $CH_4$ mixing ratio (RMT-200, Los Gatos Research, USA). The heated 6 m inlet tubes for the gas analysers were mounted 0.3 m below the centre of the anemometer sound paths. The inner tube diameter and the flow rate were 3.1 mm and 6 l min$^{-1}$, and 8 mm and 15 l min$^{-1}$ for the LI-7000 and RMT-200, respectively.

The EC data were sampled at 10 Hz, and standard methods were used to calculate half-hourly turbulent fluxes (Aubinet et al., 2012). Block averaging and a double rotation of the coordinate system were applied first (McMillen, 1988). Water vapour fluctuations affecting $CO_2$ mixing ratios were compensated for (Webb et al., 1980), but a similar procedure was not necessary for temperature (Rannik et al., 1997). Flux losses due to high-frequency signal attenuation were corrected for using methods detailed by Moore (1986) and Tuovinen et al. (1998).

The half-hourly averaged data were screened, and the data were accepted on the basis of the following criteria: relative stationarity < 100% (Foken and Wichura, 1996), number of recorded data per 30 min > 17400, number of signal spikes per 30 min < 360, mean $CO_2$ mixing ratio = 340–550 ppm. A western wind direction sector (260°–315°), within which the ecosystem changed from fen to pine forest at a distance of 100 m, was excluded. In addition, periods of insufficient turbulence were discarded with the friction velocity limit of 0.1 m s$^{-1}$.

**2.2.3 Abiotic and biotic environmental measurements**

Additional meteorological variables measured close to the EC tower included air temperature and humidity at 3 m height (Vaisala HMP 230), global and reflected radiation (Kipp&Zonen CM7), and downward and upward photosynthetic photon flux density (Kipp&Zonen PQS 1). Water vapour pressure deficit (VPD) was calculated from air temperature and relative humidity according to Jones (2013). Soil temperature profiles were measured in both a
string (at -10, -30, -50, -75 and -105 cm) and a flark (at -10, -30 and -50 cm) (IKES Pt100 sensors). The data were collected continuously by data loggers as 30 min averages. A soil temperature time series was generated for each chamber plot by adopting the -10 cm flark temperatures for F, TT and SM and the -10 cm and -30 cm string temperatures for ST, and adjusting them to match the plot-specific soil temperatures on the chamber flux measurement days.

The water table level relative to the peat surface was measured from perforated tubes placed next to each chamber measurement plot. During early summer, when there were still ice lenses inside strings, the WTL of string tops was measured as the depth of an ice lens or the melt water overlaying the lens. These measurements were conducted simultaneously with the chamber flux measurements.

Plant species coverage and mean height were measured biweekly in each collar for estimating the leaf area index
(LAI). Harvested samples of different species groups (deciduous shrubs, evergreen shrubs, forbs, graminoids and mosses) were collected during peak summer 2017 in 57 sampling plots of 50 cm x 50 cm, and LAI was measured from these samples with an A4 scanner. With the help of the samples, empirical relationships between LAI and species group coverage and height were established with ordinary least squares linear regressions (Juutinen et al., 2017). Biweekly LAI was then estimated for the collars with these empirical relationships.

The phenology of the fen vegetation was also tracked utilising daily phenocamera images taken with StarDot Netcam SC 5 digital camera. The camera was placed at 3 m in a weather-proof housing on a pole facing the north, and the viewing angle of the camera was adjusted to 45°. Images were automatically taken every 30 min, and daytime images



during the growing season were used for the image analysis. The processing of the images was executed with FMIPROT, a software designed for image processing for phenological and meteorological purposes (Linkosalmi et al., 2016; Tanis et al., 2018). The Green Chromatic Coordinate (GCC) was used as a greenness index and calculated as

$$GCC = \frac{\Sigma G}{\Sigma G + \Sigma R + \Sigma B} \qquad (2)$$

where $\Sigma G$, $\Sigma R$, $\Sigma B$ are the sums of green, red and blue channel indices, respectively, of all pixels comprising a region of interest (ROI). In addition to a more general view, ROIs were defined separately for flark and string PCTs by grouping F and TT areas and SM and ST areas together, respectively. The growing season start dates were determined based on soil temperatures measured at 10 cm depth in strings and flarks, with temperatures rising over +3 °C indicating the start of the growing season. The end dates were defined in the strings from the timing of soil freezing at 10 cm depth and in flarks from the appearance of continuous snow cover seen in the daily phenocamera images.

To estimate peat bulk density and C and N concentrations for the different PCTs, peat samples were collected from 40 typified plots placed at distances of 25 to 150 m from the EC tower in cardinal, intercardinal and secondary intercardinal directions (placement of plots is described in Räsänen and Virtanen, 2019). At each plot, a sample of approximately 5 cm × 5 cm × 5 cm was cut out of the peat at 0–5 cm depth (i.e. straight under the litter layer, where plant structures are still discernible) and at 15–20 cm depth. The samples were dried for bulk density estimates, and ground using a ball mill and their C and N concentrations were analysed using a CNS-2000 analyser (LECO Corporation, Saint Joseph, MI, USA). Peat C and N content (mg cm$^{-3}$) was then calculated using the bulk density and C and N concentrations. The pH was measured in the field using a sample of water collected from the bottom of a 30 cm deep hole at each plot.

### 2.3 Partitioning and gap-filling of CO₂ fluxes

#### 2.3.1 Environmental response functions

To fill the gaps in the collected $CO_2$ flux data, both EC- and chamber-based, and to analyse the processes controlling NEE, the $CO_2$ fluxes ($F_{NEE}$) were partitioned to two opposite flux components:

$$F_{NEE} = F_{GPP} + F_R \qquad (3)$$

where $F_{GPP}$ is the negative flux due to gross primary productivity (GPP), which represents the $CO_2$ uptake by the vegetation through photosynthesis, and $F_R$ is the positive flux due to ecosystem respiration (ER), which describes the release of $CO_2$ to the atmosphere through autotrophic and heterotrophic processes.

Gaps in the $F_{NEE}$ time series were filled with parametrised values that were estimated separately for $F_{GPP}$ and $F_R$. The dependence of $F_{GPP}$ on solar radiation was parametrised by a rectangular hyperbola (e.g., Whiting, 1994):

$$F_{GPP} = \frac{PPFD \times \alpha \times GP_{max}}{PPFD \times \alpha + GP_{max}} \qquad (4)$$

where $PPFD$ is the measured photosynthetic photon flux density, $\alpha$ is the initial slope between $F_{GPP}$ and $PPFD$, and $GP_{max}$ is the theoretical maximum gross photosynthesis rate.

The respiration flux was parametrised by an exponential dependence on temperature (Lloyd and Taylor, 1994):

$$F_R = R_{10} \times e^{E_0 \left( \frac{1}{T_0} - \frac{1}{T_s - T_1} \right)} \qquad (5)$$





where $R_{10}$ is the base respiration rate at 10 °C, $E_0$ is the activation energy, $T_0 = 56.02$ K, $T_1 = 227.13$ K, and $T_s$ is the soil temperature at the 10 cm depth.

The calculations and analyses were made with the Python programming language (Python Software Foundation, version 2.7, https://www.python.org, last access: 1 October 2020) with the NumPy (http://www.numpy.org/, last access: 1 October 2020) and SciPy (http://www.scipy.org/, last access: 1 October 2020) libraries.

### 2.3.2 Gross primary productivity parametrisation for chamber-based fluxes

For each of the chamber measurement plots, the GPP flux time series was calculated over the two growing seasons by
utilising the dependence of photosynthesis rate on solar radiation. To estimate the $F_{GPP}$ time series from the data from the six and seven measurement days in 2017 and 2018, respectively, a time-invariant, PCT-specific $k = \alpha/GP_{max}$ parameter was estimated. This was done by pooling the daily data from all plots of the same PCT, including only those data sets of different shading levels in which the highest $PPFD$ exceeded 800 µmol m$^{-2}$ s$^{-1}$, and fitting the light response curve (Eq. 4) to these data. The PCT-specific k was calculated from the fitted $\alpha$ and $GP_{max}$ values as the
variance-weighted mean k discarding the fits that had a relative error greater than 100%. The k values obtained were around 0.002 (Table 2), which is a typical value for mesotrophic boreal fens (Bubier et al. 1999).

**Table 2:** PCT-specific $k = \alpha / GP_{max}$ parameter (± standard error).

| Plant community type | k | No. of daily fittings after screening (total) | No. of data points after screening (total) |
| --- | --- | --- | --- |
| String top (ST) | $0.00186 \pm 0.00057$ | 5 (8) | 47 (75) |
| String margin (SM) | $0.00262 \pm 0.00061$ | 5 (7) | 65 (84) |
| *Trichophorum* tussock (TT) | $0.00177 \pm 0.00029$ | 7 (7) | 66 (66) |
| Flark (F) | $0.00155 \pm 0.00042$ | 5 (7) | 52 (66) |

After determining the PCT-specific k values, $GP_{max}$ was estimated for each plot measurement by fitting Eq. (4) (modified to incorporate k) to the corresponding $CO_2$ flux and $PPFD$ data. $GP_{max}$ was then linearly interpolated between the measurement days assuming $GP_{max} = 0$ at the start and end of the growing season. Finally, the half-hourly $F_{GPP}$ time series were calculated for each plot from the PCT-specific k parameter and the time series of the plot-specific $GP_{max}$ parameter and $PPFD$ measurements.

**2.3.3 Respiration parametrisation for chamber-based fluxes**

A respiration flux time series was calculated for both growing seasons, separately for each PCT, by adopting the dark chamber measurements of $CO_2$ flux as respiration data and relating them to the $T_s$ measured simultaneously at the plot. The PCT-specific $E_0$ values were obtained by fitting Eq. (5) to all respiration and temperature data of each PCT (Table 3).



**Table 3:** PCT-specific activation energy ($E_0$) (± standard error).

| Plant community type | $E_0$ | No. of data points |
|---|---|---|
| String top (ST) | 273.0 ± 45.5 | 50 |
| String margin (SM) | 430.7 ± 52.8 | 65 |
| *Trichophorum* tussock (TT) | 527.9 ± 67.7 | 52 |
| Flark (F) | 1047.7 ± 101.6 | 52 |

The PCT-specific $R_{10}$ values were then calculated for each measurement day by using Eq. (5) with these $E_0$ values, respiration data and the corresponding half-hour mean soil temperatures. The chamber-based $CO_2$ flux measurements did not cover the growing season start and end days. Therefore, the $R_{10}$ values for those days were estimated with Eq. (5) by utilising the $CO_2$ fluxes measured with the EC technique during a two-week period around the growing season start and end dates. Continuous time series of half-hourly $R_{10}$ for each PCT were obtained by linear interpolation. Finally, the $F_R$ time series over the growing seasons were calculated for each plot from the PCT-specific $E_0$ parameter and the time series of the PCT-specific $R_{10}$ and the plot specific soil temperature time series.

### 2.3.4 Gap-filling of CO₂ eddy covariance fluxes

The EC flux measurement time series had gaps due to equipment failures and quality control filtering applied during the post processing of data. The gaps in the $CO_2$ flux data were filled by modelled $F_{GPP}$ and $F_R$ values, calculated with parametrised Eqs. (4) and (5), respectively, which were fitted to measurements. The fitting was performed in a moving window of at least 5 and 15 days, for $F_{GPP}$ and $F_R$, respectively, long enough to result in at least 30 half-hourly observations. The modelling was performed in two steps. First, the $F_R$ was parametrised with the nighttime data (PPFD < 30 µmol m⁻² s⁻¹), and second, by utilising obtained respiration parameters ($E_0$ and $R_{10}$) the GPP parameters ($\square$ and $GP_{max}$) were obtained by fitting $F_{NEE}$ to all available data.

### 2.4 Gap-filling of CH₄ fluxes

The CH₄ flux ($F_{CH4}$) time series collected with chambers were gap-filled separately for each plant community type by assuming an exponential temperature dependence (Kim et al., 1999):

$$F_{CH4} = a_{10} \times Q_{10}^{(T_s - T_0)/T_0} \tag{6}$$

where $a_{10}$ is the CH₄ flux at 10 °C, $Q_{10}$ is the temperature coefficient, $T_s$ is soil temperature at 10 cm depth and $T_0 =$ 10 °C. Including water table level as an additional explanatory variable to the model did not improve the model.

The $Q_{10}$ coefficient was determined for each PCT from the data set of measured daily CH₄ fluxes and soil temperatures from each plot that included both growing seasons (Table 4).

For each of the measurement days, the PCT-specific $a_{10}$ values were calculated by using Eq. (6) with the previously determined $Q_{10}$ coefficients and measured soil temperatures. For days in the growing season beginning and end, where there were no chamber measurements, the $a_{10}$ values were estimated by using the CH₄ fluxes measured with the EC technique during a two-week period in the start and end of the growing season. The $a_{10}$ data were linearly interpolated between the measurement days to obtain a continuous half-hourly time series. The CH₄ flux time series for each PCT





were calculated for both growing seasons by using the time invariant $Q_{10}$ coefficients, $a_{10}$ time series and the continuously measured soil temperature at each plot.

**Table 4:** PCT-specific $Q_{10}$ coefficient (± standard error).

| Plant community type | $Q_{10}$ | No. of data points after screening (total) |
|---|---|---|
| String top (ST) | $1.40 \pm 0.38$ | 49 (50) |
| String margin (SM) | $4.50 \pm 1.10$ | 52 (65) |
| *Trichophorum* tussock (TT) | $5.46 \pm 1.24$ | 52 (52) |
| Flark (F) | $8.33 \pm 1.86$ | 48 (52) |

For filling gaps in the filtered EC time series of $CH_4$ fluxes, a simple moving average interpolation of the half-hour fluxes was used. A moving average window of ± 1, 2, 4, 8, 16 or 32 days was used depending on the length of the gaps in data.

**2.5 Estimating flux uncertainty**

The uncertainty of the annual $CO_2$ and $CH_4$ balances obtained from the EC-based fluxes were estimated by taking account of the most significant error sources. The random error estimate included the statistical measurement error ($E_{meas}$) and the error caused by gap-filling of missing data ($E_{gap}$) (Räsänen et al., 2017):

$$E_{meas/gap} = \sqrt{\sum_i \frac{(F_{i,obs} - F_{i,mod})^2}{n_{obs}}} \sqrt{n_{obs/gap}} \qquad (7)$$

where $F_{obs}$ is the half-hourly $CH_4$ or $CO_2$ flux that remained after all the filtering procedures, $F_{mod}$ is the corresponding fitted value and $n_{obs/gap}$ is the number of observed or gap-filled data. This provides a conservative error estimate for $E_{meas}$, and for $E_{gap}$ includes the effect of random variability on the model fits (Aurela et al., 2002).

Additionally, the annual error due to friction velocity filtering ($E_{ustar}$) was estimated by recalculating the annual EC-based $CO_2$ and $CH_4$ balances with modified data sets that were screened with two additional friction velocity limits

(0.05 and 0.15 m s$^{-1}$). $E_{ustar}$ was calculated as the average deviation from the annual balance calculated with the optimal friction velocity limit (0.1 m s$^{-1}$) (Aurela et al., 2002).

The total uncertainty of the annual EC-based $CO_2$ and $CH_4$ balances was calculated as

$$E_{tot} = \sqrt{E_{meas}^2 + E_{gap}^2 + E_{ustar}^2} \qquad (8)$$

The uncertainty of the PCT-specific chamber-based fluxes $F_{GPP}$, $F_R$ and $F_{CH4}$ were estimated by combining the

uncertainty due to the estimated parameters ($GP_{max}$, $R_{10}$ and $a_{10}$) and the flux variation among the plots of each PCT. The uncertainty of $F_{NEE}$ was calculated assuming that the uncertainties of $F_{GPP}$ and $F_R$ are independent. The uncertainty estimate was calculated for each half hour in the time series and further for the monthly and growing season sums.

We assumed that random errors in the response function fitting parameters are independent, so the standard error of the function $f$ (either $F_{GPP}$, $F_R$ or $F_{CH4}$) was calculated as



$$\sigma_f = \sqrt{\left(\frac{\partial f}{\partial a}\right)^2 \sigma_a^2 + \left(\frac{\partial f}{\partial b}\right)^2 \sigma_b^2} \qquad (9)$$

where a and b are $R_{10}$ and $E_0$ for $F_R$, $GP_{max}$ and k for $F_{GPP}$ and $a_{10}$ and $Q_{10}$ for $F_{CH4}$, and $\sigma_a$ and $\sigma_b$ denote their standard errors, respectively.

### 2.6 Linear mixed-effects models

The effect of environmental variables on ER, GPP, NEE and CH$_4$ fluxes was evaluated with the linear mixed-effects
(LME) model that was fitted by maximum likelihood. The chamber flux measurement data of the 13 measurement
days from both years were used in this analysis, and all four PCTs were pooled together. Both logarithmically
transformed and non-transformed response variables were tested, and the final model was chosen based on model
residual plots. The $F_R$ data were transformed logarithmically. Normalisation of $F_R$ to 10 °C was tested, but it did not
improve the model performance. The $F_{NEE}$ and $F_{GPP}$ data were normalised to a common radiation level of PPFD =
1200 µmol m$^{-2}$ s$^{-1}$ (denoted as $F_{NEE1200}$ and $F_{GPP1200}$), which represent a near-optimal radiation level for photosynthesis
in northern ecosystems (Laurila et al., 2001). For the CH$_4$ fluxes, daily mean values were used with a logarithmic
transformation. Normalisation of $F_{CH4}$ to 10 °C was tested, but it did not improve the model.

The following fixed explanatory variables were tested in the models for $F_R$ and $F_{NEE1200}$: $T_s$, WTL, GCC and total
vascular LAI; for $F_{GPP1200}$: WTL, GCC, daily maximum VPD and vascular LAI; and for CH$_4$ flux: soil temperature (at
-10 cm), WTL, GCC and LAI divided into four plant groups (deciduous shrubs, evergreen shrubs, forbs and
graminoids). Highly cross-correlated explanatory variables (Pearson correlation coefficient r >|0.7|) were excluded
from the models (Table A4), and the number of variables for the final model was reduced with a backward stepwise
procedure by minimising Akaike's Information Criterion value. In all regressions, the measurement plot was included
as a random effect. To evaluate the relative impact of each explanatory variable, the standardised regression
coefficients were calculated, while the marginal coefficient of determination ($R_m^2$) was used to quantify how much the
fixed effects explain of the variance of the response variable. Data analyses were conducted in R (R Core Team, 2019)
with the packages nlme (Pinheiro et al., 2019), MASS (Venables and Ripley, 2002) and reghelper (Hughes, 2020).

## 3 Results

### 3.1 Environmental conditions

Compared to long-term statistics measured at the Inari Ivalo weather station, the growing season of 2017 at Kaamanen
fen had an average temperature, but high precipitation sum, while the growing season 2018 was warm and dry (Fig.
2). The annual average temperature was close to the long-term value (-0.4 °C) in both years (-0.6 °C in 2017 and 0.4
°C in 2018), but in 2018 the monthly means of May, July and November were clearly higher than the reference values,
by 4.4, 5.4 and 6.5 °C, respectively (Fig. 2a). On average, the April–October period was considerably warmer in 2018
(7.7 °C) than in 2017 (5.5 °C) (Figs. 2 and 3a). In 2018, the daily mean temperatures rose to 10 °C already in early
May, while in 2017 such temperatures were not recorded until early June. In 2018, the daily mean exceeded 20 °C on
12 days, and the maximum daily mean temperature of 25.9 °C was recorded on 18 July. In comparison, the daily mean



temperatures never rose over 20 °C during 2017, the maximum being 19.3 °C (28 July). In August, September and October, the differences between the years were not as large as in July.

The annual precipitation sum was higher in 2017 (499 mm) than the long-term average (472 mm) and the precipitation in 2018 (473 mm). However, there were large differences in the monthly precipitation sums between the years (Fig. 2b). In 2017, there was hardly any rain in May, while the precipitation sum of the summer months June–August was 32% higher than the 30-yr average. On the other hand, the precipitation sum of July 2018 was only 34 mm, which is less than half of the 30-yr average. This dry spell was followed by a rainy August and September.

A marked difference between the study years was observed in VPD (Fig. 3b), which serves as an indicator for drought events (Lindroth et al., 2007, Aurela et al., 2007). We defined drought as the period during which the daily maximum VPD exceeded 2 kPa. In 2017, this limit was not exceeded, while in 2018 it was exceeded in total on 13 days between 2 July and 1 August, with a maximum of 3.1 kPa observed on 18 July (Fig. 3b). The average daily maximum VPD during this period was 1.69 kPa, while during the same period in 2017 it was 0.94 kPa. The corresponding mean air

temperatures were 18.5 and 14.2 °C, respectively (Fig. 3a).

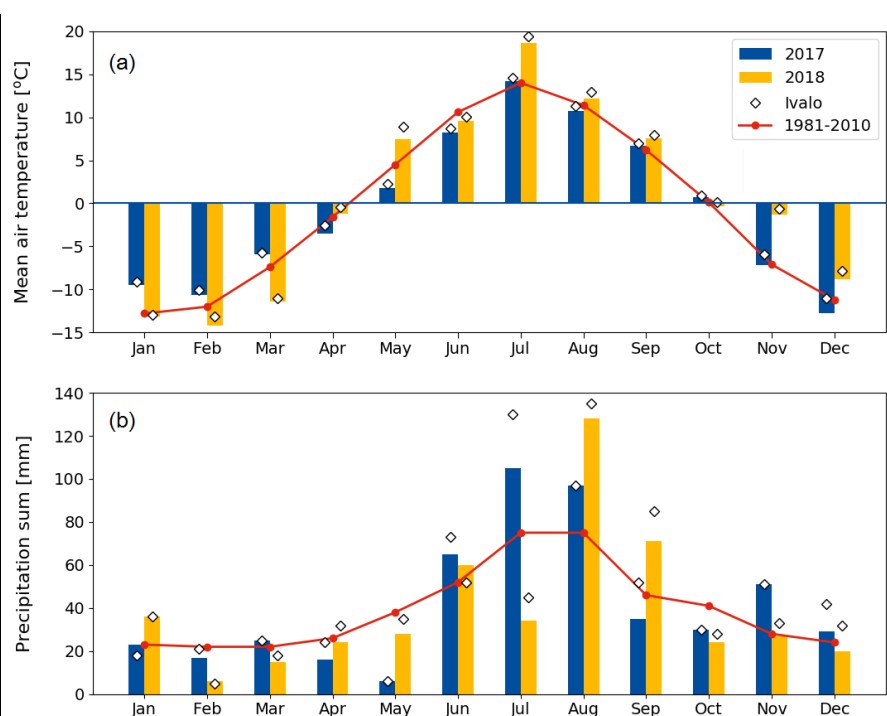

**Figure 2:** Monthly **(a)** mean air temperature and **(b)** precipitation sum in 2017 and 2018 at the Kaamanen fen, and 30-yr averages (Pirinen et al., 2012) measured at the Ivalo weather station (68° 36' N, 27° 25' E, 59 km south of Kaamanen). The monthly values

measured at Ivalo in 2017 and 2018 are marked with diamonds.

Soil temperatures (Fig. 3c) varied from day to day less than the air temperature. The temperatures measured within the ST communities were lower than those of F, TT and SM until late summer due to the presence of ice lenses in strings. In F, TT and SM, the average soil temperature during May–September was 9 °C in 2017 and 11 °C in 2018,



while in ST it was 6 °C in 2017 and 8 °C in 2018. The difference in the maximum daily soil temperatures between the
years, about 2 °C for F, TT and SM and 3 °C for ST, was not as prominent as in the air temperatures.

The growing season, as defined by the 3 °C soil temperature limit, began two weeks earlier in 2018 than in 2017 (Section 2.2.3) (Table 5). In spring, when the snow is melting, the string plant communities (SM, ST) are first exposed to direct sunlight, and therefore the growing season began a few weeks earlier in those plant communities. The growing season ended as the peat soil froze and continuous snow cover was established in October.

The greenness index GCC showed a clear seasonal pattern (Fig. 3d). The GCC variation during May – June 2017 and in May 2018 coincided with the soil temperature rise and snow melt. The flark plant communities (F and TT) with mostly sedge and moss vegetation had a lower GCC than the string plant communities (SM and ST), which had a more diverse vegetation composition (Table 1). The drought impact on vegetation was clearly visible in the field in July 2018, and accordingly a higher maximum GCC was recorded in 2017 than 2018.


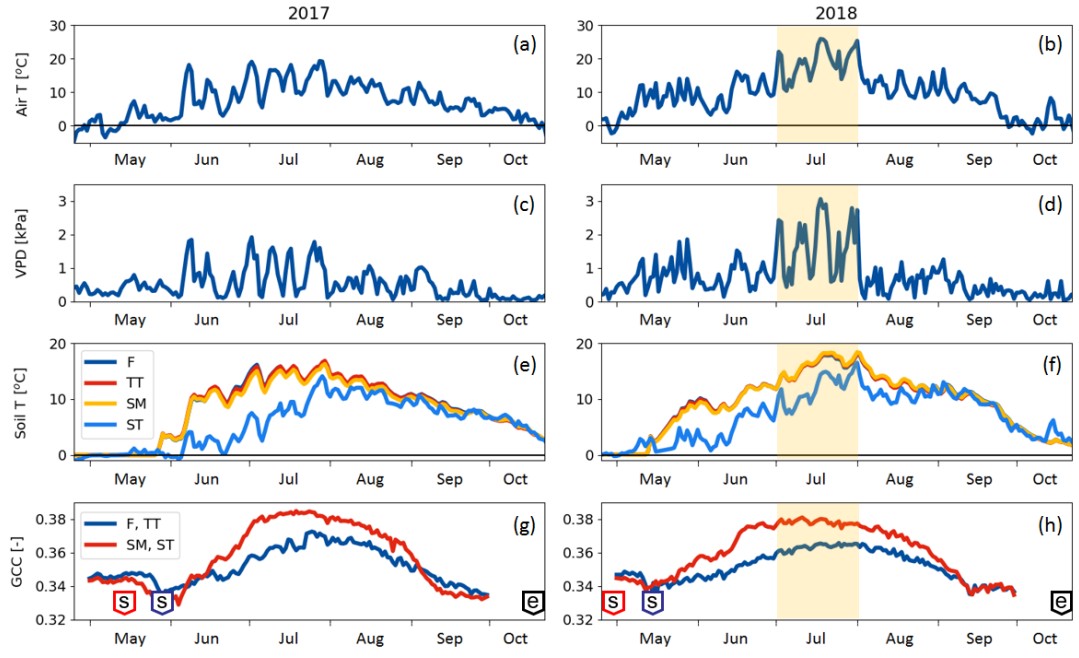

**Figure 3: (a,b)** Daily mean air temperature, **(c,d)** daily maximum vapour pressure deficit, **(e,f)** PCT-specific daily mean soil temperatures at 10 cm depth of F, TT and SM, and 20 cm depth of ST, and **(g,h)** greenness index of the flark (F, TT) and string (SM, ST) areas. The drought period (2 July – 1 August 2018) is denoted with shading. In the GCC plots **(g,h)** the **'s'** boxes indicate
the start of growing season dates and the **'e'** boxes the growing season end.

**Table 5:** Growing season start and end dates.

| Plant community type | Growing season start | Growing season end |
|---|---|---|
| F & TT | 28 May 2017 | 22 October 2017 |
|  | 14 May 2018 | 23 October 2018 |
| SM & ST | 15 May 2017 | 22 October 2017 |
|  | 27 April 2018 | 23 October 2018 |





There were significant microtopography-related differences among the PCT-specific WTL data (Fig. 4a,b). The peat

surface of flarks was usually barely submerged, while the *Trichophorum* tussocks stuck out a few centimetres from

the water. While WTL rose somewhat inside the strings, it remained at a depth of about 10 and 50 cm in the SM and

ST communities, respectively. During the early growing season, the water table depth in ST was bounded by the ice

lens depth. During July 2018, WTL dropped in all communities, which matched the drought period observed as an

increased VPD. The LAI was systematically lowest in the wettest plant community type (i.e. F) and highest in the ST

community type (Fig. 4c,d).

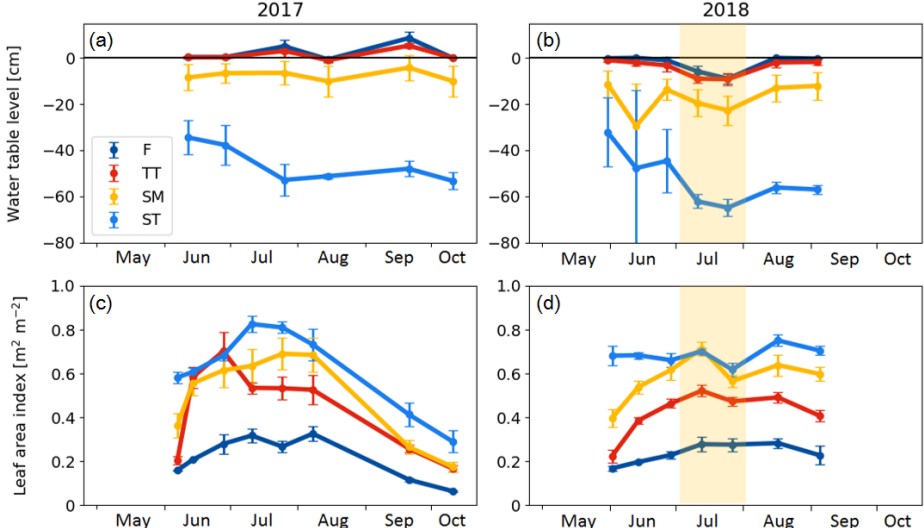


**Figure 4: (a,b)** Average plant community water table level and **(c,d)** average vascular leaf area index of the chamber plots during
2017 and 2018. Error bars represent the standard deviation within a plant community. The drought period in July 2018 is denoted
with shading.

**Table 6:** Average (± standard deviation) peat pH measured at 30 cm depth and bulk density, C and N content and C:N ratio within
the top 20 cm peat layer (measured from 0-5 cm and 15-20 cm depth) for each plant community type.

| Plant community type | pH | Bulk density [g cm⁻³] | Soil C content [mg cm⁻³] | Soil N content [mg cm⁻³] | C:N ratio | No. of sample plots |
|---|---|---|---|---|---|---|
| String top (ST) | 4.6 ± 0.4 | 0.095 ± 0.030 | 50.9 ± 15.9 | 1.5 ± 0.8 | 38.9 ± 10.5 | 20 |
| String margin (SM) | 5.7 ± 0.5 | 0.094 ± 0.046 | 44.7 ± 21.5 | 2.5 ± 1.9 | 24.3 ± 13.0 | 16 |
| *Trichophorum* tussock (TT) | 5.9 ± 0.2 | 0.133 ± 0.042 | 57.2 ± 11.3 | 3.4 ± 0.8 | 17.1 ± 1.8 | 18 |
| Flark (F) | 5.8 ± 0.2 | 0.098 ± 0.017 | 44.2 ± 8.6 | 2.7 ± 0.6 | 16.5 ± 2.2 | 18 |

### 3.2 Ecosystem-level $CO_2$ and $CH_4$ fluxes

During the winter (October–April), the ecosystem respiration determined the magnitude of $CO_2$ exchange and the fen

was a source of $CO_2$. As the spring advanced in April and May GPP gradually got started, but respiration still





dominated the NEE, and the switch to a $CO_2$ sink took place on 16 June 2017 and 30 May 2018, the difference in timing reflecting the onset of the growing season (Fig. 5 a,c,d, Table 5).

The largest difference between the years in the cumulative $CO_2$ balance was recorded in early July, when the fen had accumulated 30 g C $m^{-2}$ more in 2018 than in 2017 (Fig. 5d). This difference was generated in June, mostly due to the larger GPP in 2018. However, the increased cumulative uptake was offset later between 20 July and 9 August 2018, when GPP decreased and the fen momentarily became a $CO_2$ source at the end of the drought period (Fig. 5a).

In both years, the fen turned from $CO_2$ sink to a source in early September (Fig. 5a) even though the plants kept photosynthesising until October (Fig. 5d). After this switchover, the trajectories of the cumulative $CO_2$ balances were similar in 2017 and 2018 (Fig. 5d), leading to a small annual sink (< 10 g C $m^{-2}$) in both years (Table 7).

The fen acted as a $CH_4$ source throughout the year (Fig. 5e,f). In winter, emissions were low (0.006 g C $m^{-2}$ $d^{-1}$), after which, a distinctive $CH_4$ emission pulse was observed in May of both years. This pulse lasted about one week and 420 took place two weeks earlier in 2018 than 2017 due to the different time of the snow melt (Fig. 5e).

The drought period in July 2018 reduced the $CH_4$ emissions compared to the previous year (Fig. 5e,f). However, the drought did not affect the $CH_4$ fluxes until 22 July, i.e. 21 days since the drought started. The reduction in $CH_4$ emissions continued until the end of August, when the drought was already over.

The annual carbon balance of the fen, estimated with the EC measurements of $CO_2$ and $CH_4$ fluxes, was close to zero 425 in both years (Table 7).

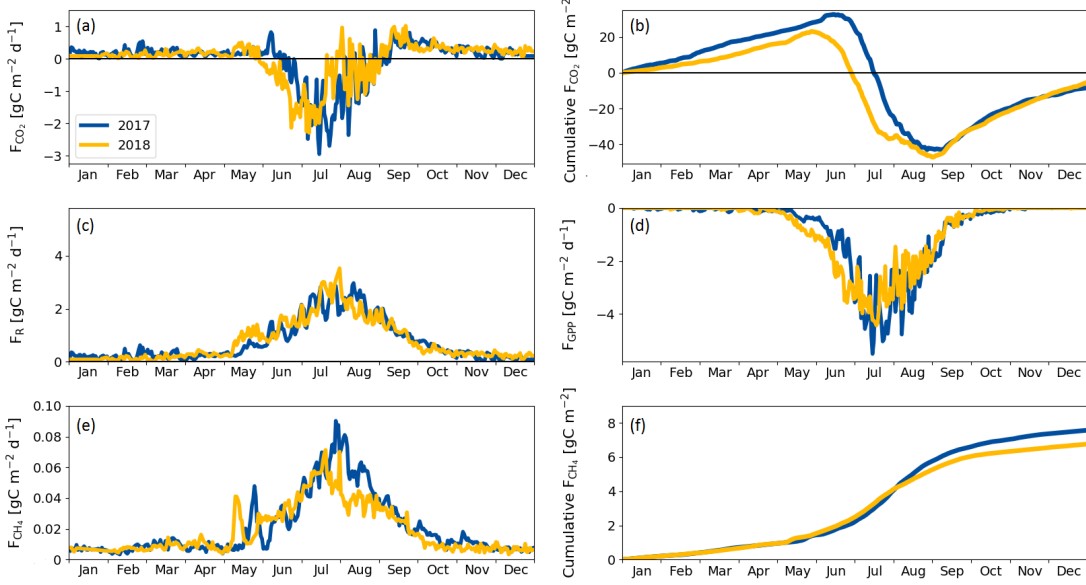

**Figure 5: (a)** Daily $CO_2$ flux, **(b)** cumulative $CO_2$ flux, **(c)** respiration flux, **(d)** gross primary productivity, **(e)** $CH_4$ flux and **(f)** cumulative $CH_4$ flux measured with the eddy covariance technique during 2017 and 2018. Negative values denote an ecosystem sink and positive values denote a flux to the atmosphere.






**Table 7:** Annual $CO_2$, $CH_4$ and carbon ($CO_2 + CH_4$) balances measured with the eddy covariance technique in 2017 and 2018.

| Year | Annual $CO_2$ balance [g C m$^{-2}$] | Annual $CH_4$ balance [g C m$^{-2}$] | Annual C balance [g C m$^{-2}$] |
|------|---------------------|---------------------|---------------------|
| 2017 | -8.5 ± 4.0 | 7.3 ± 0.2 | -1.2 ± 4.0 |
| 2018 | -5.6 ± 3.7 | 6.2 ± 0.1 | 0.6 ± 3.7 |

### 3.3 Factors affecting $CO_2$ and $CH_4$ exchange

According to the LME models, the main environmental factors controlling the C exchange at the fen were WTL, GCC and vascular LAI for $F_{NEE1200}$ and $F_{GPP1200}$, i.e. the radiation-normalised $F_{NEE}$ and $F_{GPP}$ (Table 8). Increases in GCC and vascular LAI were associated with larger $CO_2$ sink, and the positive regression coefficient of WTL indicated that drier growing locations were larger $CO_2$ sinks. $F_R$ increased with increasing soil temperature, GCC and vascular LAI, and reduced with increasing WTL. Higher $CH_4$ emissions were associated with wetter and warmer conditions, and

higher GCC and graminoid LAI. Only the total vascular LAI was needed to explain the variation in $CO_2$ flux components, while for the $CH_4$ flux LAI had to be partitioned, because the graminoid and forb LAIs showed an opposite effect on the flux.

Soil temperature was highly correlated with air temperature ($R_m^2$=0.88) and VPD ($R_m^2$=0.78), GCC was highly correlated with air temperature ($R_m^2$=0.76) and VPD ($R_m^2$=0.74), while WTL was highly correlated with evergreen

LAI ($R_m^2$=0.75) (Table A4).

**Table 8:** Standardised regression coefficients (± standard error) for the explanatory variables produced by linear mixed effect models for chamber-based $F_{NEE1200}$, $F_{GPP1200}$, $F_R$ and $F_{CH4}$.

| $F_{NEE1200}$ ($R_m^2 = 0.34$) | Regression coefficient | *p*-value |
|------|------|------|
| (Intercept) | -0.0004 ± 0.067 | 0.9949 |
| WTL | 0.185 ± 0.071 | 0.0102 |
| GCC | -0.381 ± 0.070 | < 0.0001 |
| Vascular LAI | -0.142 ± 0.080 | 0.0801 |
| $F_{GPP1200}$ ($R_m^2 = 0.64$) | | |
| (Intercept) | -0.001 ± 0.076 | 0.9928 |
| WTL | 0.269 ± 0.062 | < 0.0001 |
| GCC | -0.529 ± 0.046 | < 0.0001 |
| Vascular LAI | -0.136 ± 0.059 | 0.0209 |
| $F_R$ ($R_m^2 = 0.62$) | | |
| (Intercept) | -2.966 ± 0.094 | < 0.0001 |
| Soil temperature | 0.342 ± 0.045 | < 0.0001 |
| GCC | 0.156 ± 0.053 | 0.0034 |
| Vascular LAI | 0.300 ± 0.051 | < 0.0001 |




| | | |
|---|---|---|
| WTL | $-0.177 \pm 0.061$ | $0.0042$ |
| $F_{CH4}$ ($R_m^2 = 0.58$) | | |
| (Intercept) | $-0.692 \pm 0.093$ | $< 0.0001$ |
| Soil temperature | $0.325 \pm 0.050$ | $< 0.0001$ |
| GCC | $0.158 \pm 0.055$ | $0.0048$ |
| Graminoid LAI | $0.194 \pm 0.054$ | $0.0004$ |
| Forb LAI | $-0.169 \pm 0.053$ | $0.0018$ |
| WTL | $0.194 \pm 0.068$ | $0.0048$ |

**3.4 Plant community level $CO_2$ and $CH_4$ exchange**

The growing season ecosystem respiration sums varied among the PCTs (Fig. 6a, Table A1): it was smallest in flarks and increased with the microtopographical altitude of the PCT. The growing season respiration sums differed significantly between the years, with the exception of SM (Fig. 6a), being higher in 2018 than 2017 in the F (by 52 g C m$^{-2}$), TT (by 79 g C m$^{-2}$) and ST (by 109 g C m$^{-2}$) communities.

Similarly to ER, the GPP sums increased gradually from the wettest F to the dry SM and ST communities (Fig. 6b,
Table A1). The growing season GPP sum increased significantly from 2017 to 2018 in the F (by 33 g C m$^{-2}$), SM (by 108 g C m$^{-2}$) and ST (by 137 g C m$^{-2}$) communities, but not in TT.

Neither the PCTs nor the years differed significantly in their net ecosystem exchange of $CO_2$ (Fig. 6c). Additionally, the growing season NEE balances were similar in 2017 to 2018, because ER and GPP sums had a similar increase. This was also observed in the ecosystem-scale fluxes (Table 7). It appears that the flark communities F and TT shifted
towards being a $CO_2$ source to the atmosphere, while the string communities SM and ST shifted towards being a sink of $CO_2$.

The ST communities had distinctly the lowest $CH_4$ emission. The growing season $CH_4$ balance of the F, TT and SM communities were similar in 2018, but in 2017 SM had a slightly larger $CH_4$ emission than F and TT (Fig. 6d, Table A1). $CH_4$ emissions differed significantly between the years only in the F communities, where the growing season
balance was 2.2 g C m$^{-2}$ higher in 2018 than in 2017.

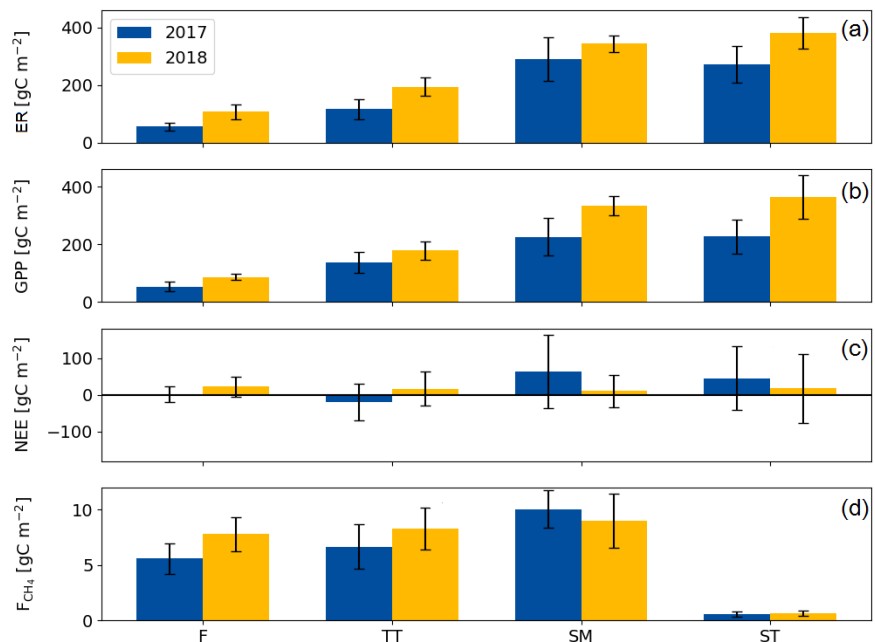

**Figure 6:** Growing season flux sums of **(a)** ecosystem respiration, **(b)** gross primary productivity, **(c)** net ecosystem exchange and **(d)** $CH_4$. Negative NEE denotes an ecosystem $CO_2$ sink. Error bars represent the 95% confidence intervals.

Seasonal variation in the ER, GPP and $CH_4$ fluxes was observed for all PCTs, with the maximum monthly exchange consistently taking place in July (Fig. 7). For NEE, however, the timing of the largest monthly exchange varied among the PCTs.

    With the exception of October, the monthly mean respiration rates were higher in 2018 than 2017, even though the difference for the TT, SM and ST communities was not significant (at the 5% significance level) during the mid-
growing season (Fig. 7a,b, Table A2). The monthly mean GPP rates in the early growing season (May–June, and for the F community also in July) were also higher in 2018 than 2017 for all PCTs (Fig. 7c,d, Table A2).

    The difference between the years was less clear in the monthly NEE. No significant differences were observed for F, whereas the NEE of TT and SM was higher in 2018 in May and April, respectively. In ST, a higher net emission could be detected in April and September 2018 and a higher net uptake in June and October 2018 (Fig. 7e,f, Table A2).

The differences in the monthly $CH_4$ emissions between 2017 and 2018 were not significant (Fig. 7g,h, Table A2). However, it seems that, while the flark plant communities F and TT had higher $CH_4$ emissions in 2018 than 2017 in the first half of the growing season (May–July), the SM community had a lower $CH_4$ emission in 2018 than 2017 in the latter part of the season (August–October).

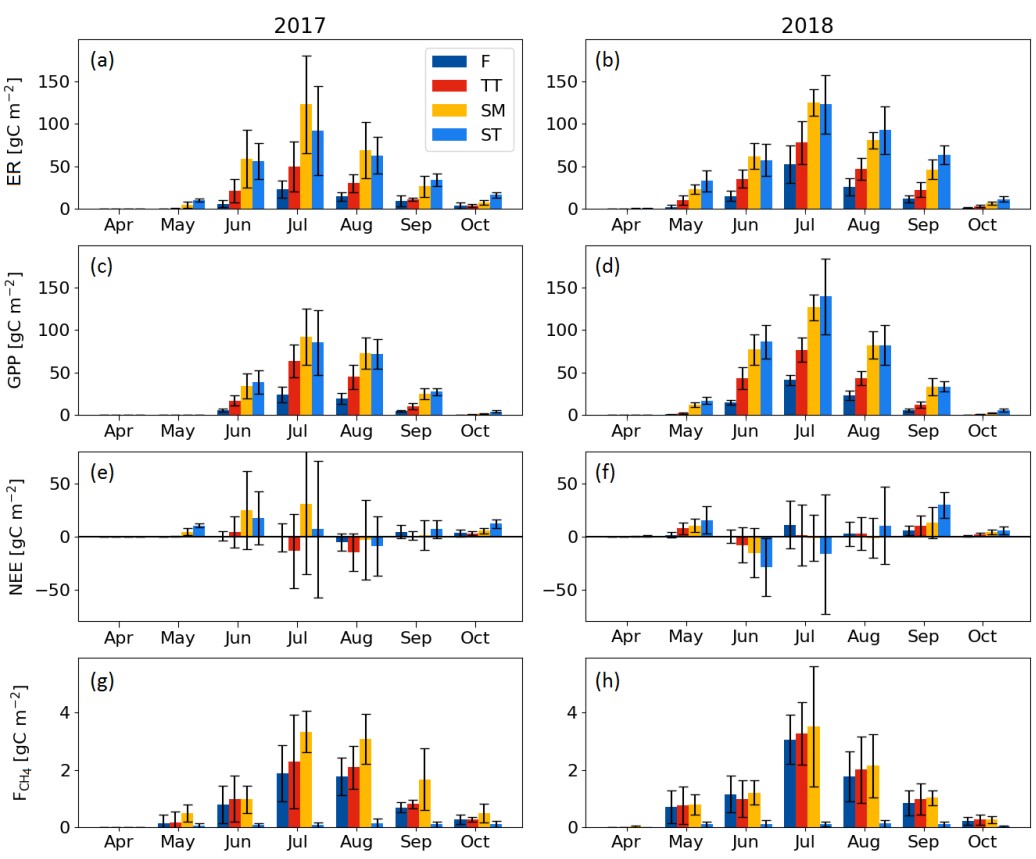


**Figure 7:** Monthly flux sums of **(a,b)** ecosystem respiration, **(c,d)** gross primary productivity, **(e,f)** net ecosystem exchange and **(g,h)** $CH_4$. Negative NEE denotes an ecosystem $CO_2$ sink. Error bars represent the 95% confidence intervals.

## 4 Discussion

Using the C flux measurements with the EC technique in 2017 and 2018, we conclude that there were two phenomena

that had a substantial effect on the annual C balance of the Kaamanen fen: the two-week difference in the growing

season start and the one-month-long drought period in 2018. Additionally, with the chamber measurements we could

capture the variation in C exchange among different plant communities.

### 4.1 Spatial variation of C exchange among plant communities

Vegetation composition of plant communities is adapted to the prevailing moisture and nutrient conditions, which can

vary greatly across the fen microtopography (Tables 1 and 6, Fig. 4). The separation of string tops from the water

table is reflected in their shrub-dominated plant community composition, which in turn creates a soil of low pH and

high C:N ratio. The ST PCT had the lowest pH of 4.6 (Table 6), i.e. a value approaching those found in bogs (pH <

4.2), while the three other PCTs had a pH (5.7–5.9) typical of mesotrophic fens (Wieder et al., 2006). Similarly, the



C:N ratio was higher in the string than flark communities. Thus, the strings can be described as nutrient-poor bog-like

islands within a mesotrophic fen (Aurela et al., 1998; Maanavilja et al., 2011). The string plant communities (SM, ST) with woody ericaceous shrubs had a relatively large LAI while the flark communities dominated by graminoids (F, TT) had a smaller LAI (Fig. 4 c,d). As expected, we found that GPP closely followed the quantity of photosynthesising plant material (Figs. 6 and 7, Table A3) (e.g. Alm et al., 1999; Bubier et al., 2003; Munir et al., 2014; Korrensalo et al., 2019). This was the case both for respiration and GPP: flarks with the highest WTL had the lowest exchange

rates, which gradually increased along with increasing height above the water table (Figs. 4 and 6a,b, Table 7). This dependence was also confirmed by statistical modelling (Table 8), and the observation is in accordance with previous findings for northern fens with microtopography (Alm et al., 1997; Strack et al., 2006; Maanavilja et al., 2011). Maanavilja et al., (2011) estimated the growing season respiration sums of the Kaamanen fen to be approximately 50, 100, 250 and 225 g C m$^{-2}$ for F, TT, SM and ST, respectively, and the growing season GPP sums to be approximately

100, 140, 290 and 250 g C m$^{-2}$ for F, TT, SM and ST, respectively, which were in general lower than our estimates (Fig. 6a,b, Table A1).

In general, the differences in NEE balances between the PCTs were less clear than those in ER and GPP (Fig. 6c, Tables A1 and A3). NEE balances have earlier been found to vary substantially in mires with a hummock-hollow microtopography, with either hollow (Maanavilja et al., 2011; Schneider et al., 2012) or hummock (Bubier et al., 2003;

Riutta et al., 2007) communities being larger net $CO_2$ sinks, or all communities being small $CO_2$ sinks (Strack et al., 2006). Maanavilja et al. (2011) estimated that in the Kaamanen fen the ST communities acted as the smallest growing season $CO_2$ sink (-10 g C m$^{-2}$), while F, TT and SM were fairly similar sinks (-50, -40 and -40 g C m$^{-2}$, respectively). This differs from our finding that in 2017 the $CO_2$ balances ranged from -20 g C m$^{-2}$ (TT) to 64 g C m$^{-2}$ (SM), while in 2018 all PCTs were small $CO_2$ sources (10–22 g C m$^{-2}$), which could be due to differing meteorological conditions.

During the measurements of Maanavilja et al. (2011) in 2007, the growing season was not as rainy as in 2017 nor had drought events similar to 2018, and the monthly air temperatures and precipitation sums were close to the 30-yr averages (Fig. 2).

In our data, $CH_4$ emissions were significantly higher from F, TT and SM than ST. In F and TT, the high emissions can be explained by the anoxic conditions that derive from the high WTL and are favourable for $CH_4$ production by

archaea as well as by the graminoids of these communities that allow effective $CH_4$ transfer to the atmosphere (Ward et al., 2013). In SM, WTL is not as high, and a possible explanation for its high $CH_4$ emission is the higher plant biomass, which provides higher substrate availability for $CH_4$ production (Korrensalo et al., 2018). The low $CH_4$ emissions from the ST communities reflect the low WTL, inefficient $CH_4$ production and the lack of $CH_4$ transport routes through graminoid plants, which leads into higher $CH_4$ oxidation (Saarnio et al., 1997, Marushchak et al., 2016).

Heikkinen et al. (2002) estimated that the monthly average $CH_4$ fluxes at this site during mid-June – end of September 1995 were 2.3 g C m$^{-2}$ on flarks, 2.0 g C m$^{-2}$ on lawns (corresponding to TT) and 0.18 g C m$^{-2}$ on strings. These values are at the high end of the range of our estimates for F (mean 1.5, range 0.7–3.1 g C m$^{-2}$ month$^{-1}$) and TT (mean 1.7, range 0.8–3.3 g C m$^{-2}$ month$^{-1}$) for the same months. Heikkinen et al. (2002) did not report high $CH_4$ emissions for SM (mean 2.1, range 1.0–3.5 g C m$^{-2}$ month$^{-1}$ in the present study), most likely due to a different classification of

string communities and lacking measurements from the margins. Our data for string tops (mean 0.11, range 0.08–0.14





g C m$^{-2}$ month$^{-1}$) is closer to the string emissions measured by Heikkinen et al. (2002). The vastly differing emissions from SM and ST suggest that the division between string margin and top is important for correctly estimating the ecosystem-scale CH$_4$ emissions as both PCTs cover a notable area of the fen (Table 1).

### 4.2 Temporal variation of C exchange

**4.2.1 Annual variation**

We estimated that the annual C balance of our fen was similar in 2017 and 2018 (Table 7), even though the meteorological conditions differed considerably between the years. The effect of an earlier onset of growing season in 2018 on the CO$_2$ balance was counterbalanced by the drought event later in that year. Aurela et al. (2004) have reported a mean annual CO$_2$ balance of -22 g C m$^{-2}$ (1997–2002) for the same fen, with variation between -4 and -53

g C m$^{-2}$. Our annual CO$_2$ balances were -8.5 and -5.6 g C m$^{-2}$ in 2017 and 2018, respectively, which are on the low side compared to Aurela et al. (2004). We assume that these differences mostly arise from the timing of snow melt and spring temperatures. Hargreaves et al. (2001) estimated the annual CH$_4$ emissions of the fen to be 5.5 g C m$^{-2}$ (measurements conducted in 1995, 1997 and 1998), while our estimates of 7.3 and 6.2 g C m$^{-2}$ in 2017 and 2018, respectively, are slightly higher.

Considering the different PCTs, we found higher growing season sums of both ER and GPP in 2018 than 2017 for all PCTs (Fig. 6a,b; Table A1). However, we did not observe significant changes in the growing season NEE (Fig. 6c, Table A1). CH$_4$ emissions differed significantly between the years in F only, with the growing season balance being 2.2 g C m$^{-2}$ higher in 2018 than 2017 (Fig. 6d, Table A1).

Differences in ER, GPP and CH$_4$ emission sums between the years are in the same direction in all plant communities,

i.e. higher values in 2018 especially during the early growing season months. From July onward, however, the changes in CH$_4$ exchange (Fig. 7, Table A2) were not as uniform. For most of these differences between the years, the spring weather emerges as a potential explanation.

### 4.2.2 Effects of spring timing

The higher ER and GPP rates during the early growing season of 2018, observed at both the ecosystem (Fig. 5c,d) and

plant community (Fig. 7a,b,c,d, Table A2) level, were most likely due to the higher temperatures and earlier growing season start in April–June 2018 (Fig. 2a, Table 5), which advanced the activity of both plants and soil microbes (Jones, 2013). We observed the effect of earlier warm conditions in soil temperatures, GCC and LAI (Figs. 3e,f,g,h and 4c,d), which were in turn found to explain the variation in ER and GPP fluxes (Table 8). Warmer temperatures have been reported to increase soil respiration in boreal mires, while the impacts on GPP and NEE have varied more (Chivers et

al., 2009; Ward et al., 2013). Early snow melt and warm spring temperatures have been suggested to increase the annual net CO$_2$ uptake of northern mires (Aurela et al., 2004; Sagerfors et al., 2008), and our results support this observation: net CO$_2$ uptake increased during the early growing season at both the ecosystem (Fig. 5a) and plant community (Fig. 7e,f) scale. Of the PCTs, the string top communities showed a particularly large increase (Table A2). The springtime increase of CH$_4$ emissions started during the thaw in May with a pulse of CH$_4$ that was stored below

the snow pack (Fig. 5e), as has been commonly observed in subarctic mires (Friborg et al., 1997; Panikov and Dedysh,



2000; Hargreaves et al., 2001; Gažovič et al., 2010). The spring pulse in May, with a peak magnitude of 0.05 g C m$^{-2}$ d$^{-1}$, accounted for approximately 0.5 g C m$^{-2}$ of CH$_4$ emissions in both years, but it occurred a few weeks earlier in 2018 (Fig. 5e). This pulse occurred simultaneously with the soil temperature rise in the flarks (Fig. 3e,f). The pulses correspond to 6–7% of the annual emissions, which is within the range of 3.5–11% found in previous studies on

subarctic fens (Friborg et al., 1997; Hargreaves et al., 2001) but larger than the proportion of < 3% that Rinne et al. (2007) observed at a boreal fen. Even though the springs differed and the hollow plant communities F and TT seemed to have higher CH$_4$ emissions in May–July in 2018 than 2017 (Table A2), the differences in the monthly CH$_4$ balances between 2017 and 2018 remained small (Fig. 7g,h, Table A2). This suggests that, unlike the CO$_2$ exchange responses, the spring weather variation mostly affects the timing but not the magnitude of CH$_4$ exchange in northern mires.

### 4.2.3 Effects of summer drought

Our results show that the C exchange of the fen was significantly affected by the drought that took place in July 2018 (Fig. 3b,d). The drought was observed as a water level drawdown by 5–20 cm, higher-than-average temperatures and an elevated VPD. These anomalies likely caused drought stress in plants (Alm et al., 1999). The EC measurements suggest that this event decreased net CO$_2$ uptake by decreasing GPP rapidly (Fig. 5a,d), most likely because the plants

regulated their stomatal openings and gas exchange as water availability decreased.

The drought impact was less obvious in chamber measurements, i.e. at the PCT level, but a bit surprisingly it appears that CO$_2$ uptake decreased in the wet flark communities F and TT and increased in the dry string communities SM and ST during the drought. The CH$_4$ emissions from SM also seemed to decrease following the reduced methanogenesis in the newly formed oxic layer (Deppe et al., 2010), but this took place after the drought in August–

October (Fig. 7g,h, Table A2). In spring, a lack of precipitation would not affect water availability as much as in midsummer, as the melt water from snow maintains a high water table level. While it is known that respiration depends on WTL (Alm et al., 1997; Christensen et al., 1998; Bubier et al., 2003), it appears that the effect of WTL on temporal ER variation is specific to a mire and a microform type. For instance, Strack et al. (2006) reported a significant increase in ER during drought events on a poor fen with microtopography, while Deppe et al. (2010) found no significant effect

of WTL fluctuations on CO$_2$ exchange on an ombrotrophic bog and alpine wetland, and Aurela et al. (2007) found a nonmonotonic link between ER fluxes and WTL on a sedge fen.

The drought decreased the ecosystem-scale CH$_4$ emissions temporarily (Fig. 5e). The drought-induced decrease is likely due to the reduced volume of anoxic peat, while the oxic zone increased correspondingly, thus reducing CH$_4$ production and increasing CH$_4$ oxidation (Strack and Waddington, 2008; White et al., 2008; Deppe et al., 2010).

However, we also observed differences among the PCT-specific responses; the SM communities reacted most to the drought, with smaller August emissions in 2018 than 2017 (Fig. 7g,h, Table A2). In F, the growing season CH$_4$ emissions were higher in 2018 than 2017, most likely due to the higher soil temperatures in 2018, as CH$_4$ emissions increase with increasing peat temperature until an optimal methanogenesis temperature within 20–30 °C (e.g. Dunfield et al., 1993; Laine et al., 2007b). In the other PCTs, no systematically higher emissions were observed in 2018,

probably since the temperature-induced enhancement was offset by the lower WTL during the drought. At the ecosystem level, the difference in the annual CH$_4$ balance remained minor (7.3 vs. 6.2 g C m$^{-2}$). This indicates that the



microtopography typical to northern mires may be a factor that increases the stability of their $CH_4$ emissions with respect to abrupt environmental changes such as drought and heatwaves.

In an earlier study, the timing of snow melt was shown to be a key parameter controlling the annual $CO_2$ balance of

the Kaamanen fen (Aurela et al., 2004). In our study, however, the higher C sequestration during the longer growing season in 2018 was offset by the drought in July. This indicates that extreme weather events can substantially affect the annual C balance of northern fens through affecting their $CO_2$ and $CH_4$ exchange. However, the drought, which covered the whole north-western Europe, did not affect the $CO_2$ and $CH_4$ exchange at Kaamanen as much as it did at more southern mires during the same period (Rinne et al., 2020). The water availability of fens that have more or less

continuous flow through them have greater resilience to water level drawdown during droughts than ombotrophic bogs. Also, here the fen microtopography seems to increase the resistance to C loss at the ecosystem scale, as each of the plant communities are adapted to different environmental conditions. This could be seen with the string plant communities that acted as $CO_2$ sinks during the drought, thus decreasing the overall drought impact on the fen, which most likely is a direct consequence of string top species being already adapted to drier environments.

**5 Conclusions**

We studied $CO_2$ and $CH_4$ exchange of a subarctic fen and found both sensitivity and inherent resilience in their response to meteorological variation. Even though meteorological and environmental conditions differed in many ways between the two measurement years, our EC data showed that the annual C balance of the fen did not differ markedly between the years. While the relatively early onset of the growing season in 2018 strengthened the $CO_2$ sink,

this gain was counterbalanced by a later drought period. Variations in water table level, soil temperature and vegetation characteristics (leaf area and greenness) explained the majority of the variation in the ecosystem-level C exchange. These environmental factors also varied among the PCTs, which was reflected in their widely differing $CO_2$ and $CH_4$ fluxes. The flark and *Trichophorum* tussock communities had lower ER and GPP than string margins and tops. Even though the string margin and top PCTs were similar in terms of $CO_2$ exchange, $CH_4$ emissions from string margins

were notably larger than those from string tops. In 2017, they even clearly exceeded the emissions from flarks and *Trichophorum* tussocks. The mean ER and GPP fluxes of all PCTs were higher during the warmer 2018 growing season than in 2017, while the changes in NEE and $CH_4$ fluxes were lesser.

The characteristic microtopography present at the Kaamanen fen generates a wide range of environmental conditions. This sustains a diversity of adapted plant and microbe communities, making the fen resilient to C loss during extreme

meteorological events. However, if drought events become more common, the long-term impacts on ecosystem functioning may be more drastic than what was observed in our study.





## Appendix A: Statistical tests and analysis

**Table A1:** PCT-specific growing season sums of ER, GPP, NEE (= ER – GPP) and CH$_4$ flux. Statistically significant differences between 2017 and 2018 are indicated with an asterisk (Z test, P < 0.05).

| PCT | 2017 | 2018 | Change |
|---|---|---|---|
| ER [g C m$^{-2}$] | | | |
| F | 56.4 ± 13.6 | 108.5 ± 25.8 | 52.1 * |
| TT | 116.5 ± 34.3 | 195.0 ± 32.5 | 78.5 * |
| SM | 289.2 ± 75.4 | 343.7 ± 28.1 | 54.5 |
| ST | 272.3 ± 62.3 | 381.1 ± 54.8 | 108.9 * |
| GPP [g C m$^{-2}$] | | | |
| F | 53.9 ± 16.6 | 86.5 ± 10.0 | 32.6 * |
| TT | 136.4 ± 36.7 | 178.2 ± 32.1 | 41.8 |
| SM | 225.4 ± 65.2 | 333.4 ± 33.7 | 108.0 * |
| ST | 226.7 ± 59.0 | 363.8 ± 74.8 | 137.1 * |
| NEE [g C m$^{-2}$] | | | |
| F | 2.6 ± 21.5 | 22.0 ± 27.7 | 19.4 |
| TT | -19.9 ± 50.2 | 16.8 ± 45.7 | 36.7 |
| SM | 63.8 ± 99.7 | 10.3 ± 43.9 | -53.5 |
| ST | 45.5 ± 85.8 | 17.2 ± 92.7 | -28.3 |
| CH$_4$ exchange [g C m$^{-2}$] | | | |
| F | 5.6 ± 1.4 | 7.8 ± 1.6 | 2.2 * |
| TT | 6.7 ± 2.0 | 8.3 ± 1.9 | 1.6 |
| SM | 10.1 ± 1.7 | 9.0 ± 2.4 | -1.1 |
| ST | 0.6 ± 0.3 | 0.6 ± 0.2 | 0.0 |








**Table A2:** PCT-specific monthly ER, GPP, NEE and CH₄ flux sums. Statistically significant differences between 2017 and 2018 are indicated with an asterisk (Z test, P < 0.05).

| | | ER [g C m⁻²] | | | GPP [g C m⁻²] | | | NEE [g C m⁻²] | | | CH₄ exchange [g C m⁻²] | | |
|---|---|---|---|---|---|---|---|---|---|---|---|---|---|
| | | 2017 | 2018 | Change | 2017 | 2018 | Change | 2017 | 2018 | Change | 2017 | 2018 | Change |
| **F** | April | | | | | | | | | | | | |
| | May | 0.1 ± 0.1 | 2.1 ± 2.6 | 2.0 | 0 | 0.6 ± 0.3 | 0.6 * | 0.1 ± 0.1 | 1.5 ± 2.6 | 1.4 | 0.15 ± 0.29 | 0.71 ± 0.57 | 0.56 |
| | June | 6.1 ± 4.0 | 15.0 ± 5.8 | 8.9 * | 5.3 ± 1.9 | 14.9 ± 2.7 | 9.6 * | 0.7 ± 4.4 | 0.1 ± 6.4 | -0.6 | 0.79 ± 0.65 | 1.16 ± 0.63 | 0.37 |
| | July | 23.0 ± 9.8 | 52.4 ± 21.8 | 29.5 * | 23.9 ± 9.1 | 41.4 ± 6.0 | 17.5 * | -0.9 ± 13.4 | 11.1 ± 22.6 | 12.0 | 1.88 ± 0.98 | 3.06 ± 0.86 | 1.18 |
| | August | 14.5 ± 5.1 | 25.9 ± 10.3 | 11.4 | 19.6 ± 6.4 | 23.3 ± 5.6 | 3.7 | -5.1 ± 8.1 | 2.6 ± 11.7 | 7.6 | 1.78 ± 0.64 | 1.78 ± 0.86 | 0.00 |
| | September | 9.1 ± 6.2 | 11.8 ± 4.0 | 2.6 | 4.8 ± 1.1 | 5.9 ± 1.6 | 1.1 | 4.3 ± 6.3 | 5.9 ± 4.2 | 1.6 | 0.70 ± 0.17 | 0.86 ± 0.44 | 0.16 |
| | October | 3.7 ± 3.5 | 1.4 ± 0.7 | -2.3 | 0.3 ± 0.1 | 0.4 ± 0.1 | 0.1 | 3.4 ± 3.5 | 0.9 ± 0.7 | -2.5 | 0.28 ± 0.16 | 0.22 ± 0.13 | -0.06 |
| **TT** | April | | | | | | | | | | | | |
| | May | 0.3 ± 0.4 | 9.9 ± 5.6 | 9.6 * | 0 | 2.2 ± 0.8 | 2.2 * | 0.3 ± 0.4 | 7.7 ± 5.7 | 7.4 * | 0.18 ± 0.37 | 0.77 ± 0.66 | 0.59 |
| | June | 21.3 ± 13.6 | 35.1 ± 10.6 | 13.8 | 17.1 ± 5.7 | 43.1 ± 12.7 | 25.9 * | 4.2 ± 14.8 | -8.0 ± 16.6 | -12.2 | 1.00 ± 0.80 | 1.00 ± 0.65 | 0.00 |
| | July | 49.6 ± 29.4 | 77.9 ± 25.1 | 28.3 | 63.3 ± 19.3 | 76.5 ± 14.1 | 13.2 | -13.7 ± 35.1 | 1.4 ± 28.8 | 15.1 | 2.30 ± 1.63 | 3.27 ± 1.09 | 0.97 |
| | August | 30.0 ± 10.8 | 46.5 ± 12.9 | 16.5 | 44.7 ± 13.9 | 43.6 ± 8.3 | -1.1 | -14.7 ± 17.7 | 2.9 ± 15.3 | 17.6 | 2.09 ± 0.75 | 2.01 ± 1.15 | -0.08 |
| | September | 11.4 ± 1.8 | 22.4 ± 8.8 | 11.0 * | 10.6 ± 3.6 | 11.9 ± 3.7 | 1.3 | 0.9 ± 4.0 | 10.5 ± 9.6 | 9.6 | 0.81 ± 0.15 | 0.99 ± 0.54 | 0.17 |
| | October | 3.8 ± 1.9 | 3.1 ± 1.4 | -0.7 | 0.6 ± 0.2 | 0.9 ± 0.2 | 0.3 | 3.2 ± 1.9 | 2.3 ± 1.5 | -0.9 | 0.28 ± 0.09 | 0.27 ± 0.17 | -0.01 |
| **SM** | April | 0 | 0.4 ± 0.3 | 0.4 * | | | | 0 | 0.4 ± 0.3 | 0.4 * | 0 | 0.03 ± 0.04 | 0.03 |
| | May | 4.6 ± 3.6 | 23.0 ± 5.4 | 18.4 * | 0 | 12.5 ± 2.7 | 12.5 * | 4.6 ± 3.6 | 10.6 ± 6.0 | 6.0 | 0.50 ± 0.30 | 0.80 ± 0.35 | 0.30 |
| | June | 58.9 ± 33.9 | 61.7 ± 15.2 | 2.8 | 34.2 ± 14.5 | 77.0 ± 17.8 | 42.8 * | 24.7 ± 36.9 | -15.3 ± 23.4 | -40 | 0.98 ± 0.47 | 1.21 ± 0.43 | 0.24 |
| | July | 122.7 ± 57.1 | 125.1 ± 15.8 | 2.4 | 91.9 ± 33.5 | 126.5 ± 15.3 | 34.6 | 30.8 ± 66.2 | -1.4 ± 21.9 | -32.2 | 3.34 ± 0.72 | 3.51 ± 2.09 | 0.18 |
| | August | 69.3 ± 33.2 | 80.7 ± 9.6 | 11.4 | 72.6 ± 18.1 | 82.1 ± 16.0 | 9.5 | -3.3 ± 37.8 | -1.4 ± 18.7 | 1.9 | 3.08 ± 0.87 | 2.15 ± 1.10 | -0.93 |
| | September | 26.3 ± 12.7 | 46.2 ± 11.3 | 19.9 * | 25.0 ± 6.5 | 33.2 ± 9.8 | 8.2 | 1.3 ± 14.3 | 13.1 ± 15.0 | 11.7 | 1.68 ± 1.07 | 1.03 ± 0.25 | -0.65 |
| | October | 7.3 ± 2.6 | 6.5 ± 2.2 | -0.8 | 1.7 ± 0.7 | 2.2 ± 0.5 | 0.5 | 5.6 ± 2.6 | 4.4 ± 2.2 | -1.2 | 0.49 ± 0.32 | 0.27 ± 0.11 | -0.23 |
| **ST** | April | 0 | 0.7 ± 0.6 | 0.7 * | | | | 0 | 0.7 ± 0.3 | 0.7 * | 0 | 0.00 ± 0.01 | 0 |
| | May | 10.4 ± 1.7 | 32.7 ± 12.3 | 22.3 * | 0 | 17.1 ± 4.0 | 17.1 * | 10.4 ± 1.7 | 15.7 ± 12.9 | 5.3 | 0.06 ± 0.08 | 0.10 ± 0.09 | 0.05 |
| | June | 56.2 ± 21.1 | 57.4 ± 18.8 | 1.2 | 38.4 ± 13.7 | 86.0 ± 19.9 | 47.7 * | 17.8 ± 25.2 | -28.7 ± 27.4 | -46.5 * | 0.08 ± 0.06 | 0.11 ± 0.13 | 0.03 |
| | July | 92.2 ± 52.3 | 122.9 ± 34.4 | 30.7 | 85.1 ± 37.7 | 139.5 ± 44.6 | 54.4 | 7.1 ± 64.5 | -16.6 ± 56.3 | -23.7 | 0.08 ± 0.08 | 0.11 ± 0.09 | 0.03 |
| | August | 62.8 ± 21.6 | 92.5 ± 27.9 | 29.6 | 71.9 ± 17.6 | 81.9 ± 23.4 | 10.1 | -9.0 ± 27.9 | 10.5 ± 36.4 | 19.6 | 0.14 ± 0.15 | 0.13 ± 0.12 | -0.01 |
| | September | 34.3 ± 7.2 | 63.4 ± 11.1 | 29.1 * | 27.3 ± 4.4 | 33.6 ± 5.7 | 6.3 | 7.0 ± 8.4 | 29.8 ± 12.5 | 22.8 * | 0.11 ± 0.10 | 0.11 ± 0.09 | 0.00 |
| | October | 16.3 ± 3.6 | 11.5 ± 2.9 | -4.8 * | 4.0 ± 1.5 | 5.7 ± 1.7 | 1.6 | 12.2 ± 3.9 | 5.8 ± 3.4 | -6.4 * | 0.11 ± 0.11 | 0.04 ± 0.04 | -0.07 |





**Table A3:** Comparison of the growing season ER, GPP, NEE and CH$_4$ flux sums between the plant community types. Statistically significant differences (Z test, $P < 0.05$) are indicated with an asterisk.

| ER absolute difference [g C m$^{-2}$] | | | | | | | |
|---|---|---|---|---|---|---|---|
| 2017 | TT | SM | ST | 2018 | TT | SM | ST |
| F | 60.1 * | 232.8 * | 215.8 * | F | 86.5 * | 235.2 * | 272.6 * |
| TT | | 172.7 * | 155.8 * | TT | | 148.7 * | 186.1 * |
| SM | | | 16.9 | SM | | | 37.4 |
| **GPP absolute difference [g C m$^{-2}$]** | | | | | | | |
| 2017 | TT | SM | ST | 2018 | TT | SM | ST |
| F | 82.5 * | 171.5 * | 172.9 * | F | 91.7 * | 246.9 * | 277.3 * |
| TT | | 89.0 * | 90.3 * | TT | | 155.2 * | 185.7 * |
| SM | | | 1.3 | SM | | | 30.4 |
| **NEE absolute difference [g C m$^{-2}$]** | | | | | | | |
| 2017 | TT | SM | ST | 2018 | TT | SM | ST |
| F | 22.5 | 61.2 | 43.0 | F | 5.2 | 11.8 | 4.7 |
| TT | | 83.7 | 65.4 | TT | | 6.5 | 0.5 |
| SM | | | 18.3 | SM | | | 7.0 |
| **CH$_4$ exchange absolute difference [g C m$^{-2}$]** | | | | | | | |
| 2017 | TT | SM | ST | 2018 | TT | SM | ST |
| F | 1.1 | 4.5 * | 5.0 * | F | 0.5 | 1.2 | 7.2 * |
| TT | | 3.4 * | 6.1 * | TT | | 0.7 | 7.7 * |
| SM | | | 9.5 * | SM | | | 8.4 * |







**Table A4:** Squared correlation coefficients ($R_m^2$) between the variables included in the LME model analysis.

| | $NEE_{1200}$ | $GPP_{1200}$ | ER | $CH_4$ flux | Air T | Soil T | WTL | VPD | GCC | LAI evergreen | LAI deciduous | LAI forb | LAI graminoid | LAI vascular | Sphagnum sp. | Other moss |
|---|---|---|---|---|---|---|---|---|---|---|---|---|---|---|---|---|
| $NEE_{1200}$ | 1 | 0.87 | -0.34 | -0.12 | -0.22 | -0.25 | 0.38 | -0.14 | -0.53 | -0.26 | -0.38 | -0.53 | 0.01 | -0.49 | -0.17 | 0.01 |
| $GPP_{1200}$ | 0.87 | 1 | -0.75 | -0.18 | -0.49 | -0.41 | 0.54 | -0.41 | -0.73 | -0.43 | -0.59 | -0.66 | 0.06 | -0.70 | -0.22 | 0.02 |
| ER | -0.34 | -0.75 | 1 | 0.19 | 0.65 | 0.45 | -0.53 | 0.61 | 0.69 | 0.47 | 0.62 | 0.56 | -0.11 | 0.69 | 0.19 | -0.01 |
| $CH_4$ flux | -0.12 | -0.18 | 0.19 | 1 | 0.50 | 0.53 | 0.31 | 0.49 | 0.32 | -0.33 | 0.01 | -0.18 | 0.49 | 0.03 | 0.15 | -0.08 |
| Air T | -0.22 | -0.49 | 0.65 | 0.50 | 1 | 0.88 | -0.12 | 0.94 | 0.76 | 0.04 | 0.26 | 0.28 | 0.23 | 0.36 | 0.01 | 0.09 |
| Soil T | -0.25 | -0.41 | 0.45 | 0.53 | 0.88 | 1 | 0.05 | 0.78 | 0.66 | -0.18 | 0.20 | 0.13 | 0.40 | 0.23 | 0.06 | 0.09 |
| WTL | 0.38 | 0.54 | -0.53 | 0.31 | -0.12 | 0.05 | 1 | -0.10 | -0.31 | -0.75 | -0.33 | -0.68 | 0.61 | -0.49 | 0.04 | -0.18 |
| VPD | -0.14 | -0.41 | 0.61 | 0.49 | 0.94 | 0.78 | -0.10 | 1 | 0.74 | 0.02 | 0.22 | 0.25 | 0.22 | 0.32 | 0.00 | 0.07 |
| GCC | -0.53 | -0.73 | 0.69 | 0.32 | 0.76 | 0.66 | -0.31 | 0.74 | 1 | 0.25 | 0.55 | 0.51 | 0.09 | 0.61 | 0.18 | 0.03 |
| LAI evergreen | -0.26 | -0.43 | 0.47 | -0.33 | 0.04 | -0.18 | -0.75 | 0.02 | 0.25 | 1 | 0.32 | 0.50 | -0.59 | 0.66 | -0.16 | 0.33 |
| LAI deciduous | -0.38 | -0.59 | 0.62 | 0.01 | 0.26 | 0.20 | -0.33 | 0.22 | 0.55 | 0.32 | 1 | 0.37 | -0.08 | 0.69 | 0.51 | -0.12 |
| LAI forb | -0.53 | -0.66 | 0.56 | -0.18 | 0.28 | 0.13 | -0.68 | 0.25 | 0.51 | 0.50 | 0.37 | 1 | -0.36 | 0.56 | -0.05 | -0.01 |
| LAI graminoid | 0.01 | 0.06 | -0.11 | 0.49 | 0.23 | 0.40 | 0.61 | 0.22 | 0.09 | -0.59 | -0.08 | -0.36 | 1 | 0.08 | 0.18 | -0.27 |
| LAI vascular | -0.49 | -0.70 | 0.69 | 0.03 | 0.36 | 0.23 | -0.49 | 0.32 | 0.61 | 0.66 | 0.69 | 0.56 | 0.08 | 1 | 0.18 | 0.03 |
| Sphagnum sp. | -0.17 | -0.22 | 0.19 | 0.15 | 0.01 | 0.06 | 0.04 | 0.00 | 0.18 | -0.16 | 0.51 | -0.05 | 0.18 | 0.18 | 1 | -0.41 |
| Other moss | 0.01 | 0.02 | -0.01 | -0.08 | 0.09 | 0.09 | -0.18 | 0.07 | 0.03 | 0.33 | -0.12 | -0.01 | -0.27 | 0.03 | -0.41 | 1 |




*Data availability.* The measured flux and ancillary meteorological and environmental data are available in Zenodo
(https://doi.org/10.5281/zenodo.3965739, Heiskanen et al., 2020).

*Author contributions.* MA, TP, SJ, LH and TL designed the study. Field flux measurements and maintenance were
carried out by LH and MA. AR, TV and SJ measured and analysed the vegetation and land cover data. JM measured
and analysed the soil chemistry data. ML produced the GCC data. AR and TV conducted linear mixed-effects model
analysis. The rest of the data analysis was carried out by LH, JPT, MA and AL. LH wrote the paper with contributions
from all co-authors.

*Competing interests.* The authors declare that they have no conflict of interest.

*Acknowledgements.* We thank Jani Antila, Holtti Hakonen and Tuuli Lehtosalo for field assistance. This work was
supported by the Finnish Meteorological Institute, the University of Helsinki, the Natural Resources Institute Finland
and the Academy of Finland CAPTURE-project (Carbon dynamics across Arctic landscape gradients: past, present
and future) grant (no. 296423).



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
