# Peer review of "Carbon dioxide and methane exchange of a patterned subarctic fen during two contrasting growing seasons"

_Biogeosciences, 2020_

## Referee Comment (RC1) · Anonymous Referee #1 · 27 Nov 2020

In brief, I found paper written very well, with a nice literature review, useful figures, tables, schemes and easy-to-understand English. Methods are state of the art. Basic tools of studying greenhouse gas exchange between atmosphere and ecosystems such as Eddy Covariance and chamber method were used in a proper way.

Finland wetlands are investigated on an incredible (best all over the world) level in terms of greenhouse gas fluxes with a lot of possibilities to compare results and to provide extrapolations. In this situation it is really hard to say something new about fluxes from these subarctic mires. In general, two years are not enough for reliable estimates of weather/climate induced effects on carbon fluxes. But I think authors did everything

they can to generate new insights about carbon balance and its environmental controls. Therefore the paper definitely deserves publication.

I have several minor comments and suggestions to make paper text a bit more clear.

L. 104. Did you miss minus sign here (if you use micrometeorological sign convention)?

L. 122. Please add information, in what year(s) collars were installed on your sites. Did you notice any changes of plant communities inside "the oldest" collars? Sometimes vegetation inside the collars starts to degrade after several years after installation.

L. 126. Did you notice the diurnal dynamic of methane emission? Can it affect any results of methane flux linear modelling?

L. 300. Please add information how much data (in % of growing season length) was gap-filled in EC fluxes of carbon dioxide and methane.

L. 620. I think that you should mention that your C-balance estimate did not include dissolved and particulate carbon loss due to water flow. May be it is not that important for overall C-balance, but it is better to remind the reader about that. Probably you know papers, where information about dissolved organic carbon transport in Kaamanen fen is presented.

L. 620. Let me also ask, do you compare the methane budget for the whole Kaamanen fen based on chamber and EC data separately (using land cover map and footprint estimate)? Are they the same or there is a difference? It is important sometimes to check yourself about proper use of available methods. Potential gap between these estimates could show that we miss something important (for example ebullition in chamber flux data). It is just a recommendation of course, I understand that you have already presented enough good data.

---

## Referee Comment (RC2) · Anonymous Referee #2 · 1 Dec 2020

General comments: This is an impressive study that should be published with minor revisions. The problem addressed is important – what are northern wetlands contributing to the global atmospheric greenhouse? Heiskanen et al. have carried out a very detailed two-year carbon dioxide and methane budget study of a representative 70N wetland in Finland, that will give considerable insight into similar wetlands worldwide across the Arctic and sub-Arctic. The study is very thorough and well presented, clearly written and well illustrated. I would however suggest the addition of a brief final section on the wider applicability of the results, to explain and make explicit what the implications are for our understanding of the impact of strong future warming and climate change.

[Figure]

Specific comments: 1. The manuscript is littered with acronyms, from the abstract all the way through (TT, SM, F, LIA, etc). They are all either standard abbreviations or explained on first contact but I get very lost. Please could a table be added listing all the acronyms, and maybe reminders in the figure captions. 2. Page 2 line 46 – 'if anoxia occurs' ? – maybe better as 'where anoxia occurs'. 3. Page 3 line 90 maybe more detail on the vegetation. In particular, is it all C3? Or are there C4 plants like Atriplex species present?. 4. The temperature dependence of respiration flux is taken from Lloyd and Taylor 1994 (P7 L233), and the temperature dependence of methane flux from Kim et al 1999 (P9 L282). Are these assumptions valid? - or is there information in the present study that can add to the older work? In particular, Kim et al were looking at rather different phragmites wetlands, in temperate settings, in Nebraska (43 degrees N whereas Kaamanen is 70N), perhaps more analogous to warmer sub-tropical and tropical systems and with more C4 metabolism present. 5. Page 16 line 440. The $CO_2$ flux being the same for both graminoids and forbs. Is that assumption secure? My question relates to my earlier question about the possible presence of C4 plants? – Are there any C4 plants like Atriplex species present? (and indeed are they likely to become more common? 6. For future work it would be nice to have some isotopic data. 7. Page 23 Line 636. It would be good here to have a paragraph or two that is more speculative (or perhaps in warning): we know that the climate in the Arctic and sub-Arctic is warming fast and changing – what is going to happen? Can this very detailed study give us any pointers to what is going to happen? The work in the paper is careful and well reported, but it needs to be given its wider context – Heiskanen et all are experts – what can they tell us about where these mires are going?

---

## Author Comment (AC1) · 17 Dec 2020

In brief, I found paper written very well, with a nice literature review, useful figures, tables, schemes and easy-to-understand English. Methods are state of the art. Ba-sic tools of studying greenhouse gas exchange between atmosphere and ecosystems such as Eddy Covariance and chamber method were used in a proper way. Finland wetlands are investigated on an incredible (best all over the world) level in terms of greenhouse gas fluxes with a lot of possibilities to compare results and to pro-vide extrapolations. In this situation it is really hard to say something new about fluxes from these subarctic mires. In general, two years are not enough for reliable estimates of weather/climate induced effects on carbon fluxes. But I think authors did everything they can to generate new insights about carbon balance and its environmental controls. Therefore the paper definitely deserves publication. I have several minor comments and suggestions to make paper text a bit more clear.

L. 104. Did you miss minus sign here (if you use micrometeorological sign convention)?

- Yes, thank you. Minus sign was added.

L. 122. Please add information, in what year(s) collars were installed on your sites. Did you notice any changes of plant communities inside "the oldest" collars? Sometimes vegetation inside the collars starts to degrade after several years after installation.

- The vegetation condition inside the collars did not differ between the old and new plots. The vegetation in the old collars used in the present study was healthy and did not show any signs of degradation when we started our measurements in 2017.

This information was added to the article (new text in italics here):

"Eight of the 17 aluminium collars (60 cm x 60 cm) were installed during the first days of June 2017, during the soil thawing, to accompany the collars that were already *installed previously in 2006. The overall vegetation condition and species composition inside the old collars were checked to match the new study plots.*"

L. 126. Did you notice the diurnal dynamic of methane emission? Can it affect any results of methane flux linear modelling?

- We checked this from the eddy covariance data, and found no systematic diurnal cycle in the methane fluxes during the period with unfrozen soil (May–October) (Figs. C1 and C2). The chamber methane flux time series were gap-filled by assuming an exponential temperature dependence, so the possible diurnal dynamic would not have affected the gap-filling.

[Figure]

**Figure C1:** Monthly average diel methane flux cycle measured at the Kaamanen fen eddy covariance tower in May–October 2017.

[Figure]

**Figure C2:** Monthly average diel methane flux cycle measured at the Kaamanen fen eddy covariance tower in May–October 2018.

L. 300. Please add information how much data (in % of growing season length) was gap-filled in EC fluxes of carbon dioxide and methane.

- In total, 62% (10813/17520) and 63% (11080/17520) of the 30-min EC $CO_2$ flux data were gap-filled in 2017 and 2018, respectively. For $CH_4$ flux, the corresponding proportions were 64% (11735/17520) and 70% (12208/17520).

When taking into account only the growing season period, 46% (3553/7728) and 50% (4326/8640) of the 30-min EC $CO_2$ flux data were gap-filled during the growing seasons 2017 and 2018, respectively. For CH4 flux, the corresponding proportions were 51% (3922/7728) and 58% (4995/8640).

This information was added to the article (Section 2.3.4 and 2.4).

L. 620. I think that you should mention that your C-balance estimate did not include dissolved and particulate carbon loss due to water flow. May be it is not that important for overall C-balance, but it is better to remind the reader about that. Probably you know papers, where information about dissolved organic carbon transport in Kaamanen fen is presented.

- We added the following note about the lateral carbon flow to Section 4.2.1:

"These balances do not include the lateral aquatic transfer of dissolved organic C and particulate C through the fen ecosystem. Aurela et al. (2002) estimated, based on Sallantaus (1994) and Kortelainen et al. (1997), that the leaching of total organic carbon was 7.5 g C m$^{-2}$ yr$^{-1}$."

Aurela, M., Laurila, T. and Tuovinen, J.-P.: Annual CO2 balance of a subarctic fen in northern Europe: Importance of the wintertime efflux, Journal of Geophysical Research - Atmospheres, 107, 4607, doi:10.1029/2002JD002055, 2002.

Kortelainen, P., Saukkonen, S. and Mattson, T.: Leaching of nitrogen from forested catchments in Finland, Global Biogeochem. Cycles, 11, 627– 638, 1997.

Sallantaus, T., Response of leaching from mire ecosystems to changing climate, in The Finnish Research Programme on Climate Change, Second Progress Report, vol. 1, edited by M. Kanninen and P. Heikinheimo, pp. 291–296, Publ. Acad. Finland, Helsinki, 1994.

L. 620. Let me also ask, do you compare the methane budget for the whole Kaamanen fen based on chamber and EC data separately (using land cover map and footprint estimate)? Are they the same or there is a difference? It is important sometimes to check yourself about proper use of available methods. Potential gap between these estimates could show that we miss something important (for example ebullition in chamber flux data). It is just a recommendation of course, I understand that you have already presented enough good data.

- We understand that such an exercise would be useful and have made a preliminary effort by comparing the EC-based methane fluxes with upscaled chamber-based fluxes. This was done by upscaling the growing season chamber data to the ecosystem scale with the mapped coverage of the four main plant community types within a distance of 200 m from the EC tower (excluding the forest wind sector). This upscaling exercise did not take into account the riparian fen plant community for the chamber-based estimate; nor have we conducted a complete footprint analysis at this point.

The upscaled chamber-based $CH_4$ flux data matched the EC fluxes well during the growing season 2017. This was also the case for 2018 with the exception of the drought period in July. As stated in the article, the drought impact was less obvious in the plant community level fluxes than in the EC data. The discrepancy could be due to the missing riparian fen data or the uncertainty in the estimated temperature dependence in heatwave conditions. For $CO_2$ flux, the seasonal courses were consistent, but the chamber-based growing season balance was lower than the corresponding EC-based balance. To better understand the compatibility between the measurement techniques, a more thorough comparison involving footprint weighting of the chamber data would be needed but is beyond the scope of the present study.

---

## Author Comment (AC2) · 17 Dec 2020

-We thank the reviewers for their positive and constructive comments. Please see below our response to each of the comments.

Lauri Heiskanen, Juha-Pekka Tuovinen, Aleksi Räsänen, Tarmo Virtanen, Sari Juutinen, Annalea Lohila, Timo Penttilä, Maiju Linkosalmi, Juha Mikola, Tuomas Laurila and Mika Aurela

Anonymous Referee #2

General comments:

This is an impressive study that should be published with minor revisions. The problem addressed is important – what are northern wetlands contributing to the global atmospheric greenhouse? Heiskanen et al. have carried out a very detailed two-year carbon dioxide and methane budget study of a representative 70N wetland in Finland that will give considerable insight into similar wetlands worldwide across the Arctic and sub-Arctic. The study is very thorough and well presented, clearly written and well-illustrated. I would however suggest the addition of a brief final section on the wider applicability of the results, to explain and make explicit what the implications are for our understanding of the impact of strong future warming and climate change.

Specific comments:

1. The manuscript is littered with acronyms, from the abstract all the way through (TT, SM, F, LAI, etc). They are all either standard abbreviations or explained on first contact but I get very lost. Please could a table be added listing all the acronyms, and maybe reminders in the figure captions.

- This is a good point. We added to the appendix a table listing the abbreviations (Table A1). We also included additional acronym definitions throughout the article and figure captions.

Table A1: List of abbreviations.

| Abbreviation | Definition |
|---|---|
| EC | Eddy covariance |
| ER | Ecosystem respiration |
| F | Flark |
| GCC | Green chromatic coordinate |
| GPP | Gross primary productivity |
| LAI | Leaf area index |
| LME | Linear mixed-effects |
| NEE | Net ecosystem exchange |
| PCT | Plant community type |
| PPFD | Photosynthetic photon flux density |
| ROI | Region of interest |
| SM | String margin |
| ST | String top |
| TT | *Trichophorum* tussock |
| VPD | Vapour pressure deficit |
| WTL | Water table level |

2. Page 2 line 46 – 'if anoxia occurs'? – maybe better as 'where anoxia occurs'.

- Changed to "where anoxia occurs".

3. Page 3 line 90 maybe more detail on the vegetation. In particular, is it all C3? Or are there C4 plants like Atriplex species present?

- The dominant species are listed in Table 1. There are no C4 plants in the fen, and it seems very improbable they would occur there in near future, as they are presently not found here in any natural boreal vegetation communities.

4. The temperature dependence of respiration flux is taken from Lloyd and Taylor 1994 (P7 L233), and the temperature dependence of methane flux from Kim et al 1999 (P9 L282). Are these assumptions valid? - or is there information in the present study that can add to the older work? In particular, Kim et al were looking at rather different phragmites wetlands, in temperate settings, in Nebraska (43 degrees N whereas Kaamanen is 70N), perhaps more analogous to warmer sub-tropical and tropical systems and with more C4 metabolism present.

- The temperature dependence of respiration is modelled with an exponential (modified Arrhenius) relationship that has been shown to result in an unbiased estimate across a wide range of ecosystem types and soil temperatures (Lloyd and Taylor, 1994). Here, both the base respiration and activation energy parameters were estimated from the local data. For the $CH_4$ flux, we used a fully generic exponential function with local parameter values. No prescribed parameter values were used, and the function does not involve any assumptions about the ecosystem type. It is true that Kim et al. (1999) studied a temperate marsh, which differs in many ways from our subarctic fen. For the sake of consistency, we changed this citation to Marushchak et al. (2016), which deals with a subarctic ecosystem.

Marushchak, M. E., Friborg, T., Biasi, C., Herbst, M., Johansson, T., Kiepe, I., Liimatainen, M., Lind, S. E., Martikainen, P. J., Virtanen, T., Soegaard, H., and Shurpali, N. J.: Methane dynamics in the subarctic tundra: combining stable isotope analyses, plot- and ecosystem-scale flux measurements, Biogeosciences, 13, 597–608, https://doi.org/10.5194/bg-13-597-2016, 2016.

5. Page 16 line 440. The $CO_2$ flux being the same for both graminoids and forbs. Is that assumption secure? My question relates to my earlier question about the possible presence of C4 plants? – Are there any C4 plants like Atriplex species present? (and indeed are they likely to become more common?)

- The sentence on lines 440-442 is part of the description of the statistical modelling results and does not imply that the $CO_2$ flux would be the same for graminoids and forbs; rather, it reports the variables that explain the observed flux variation. As stated in the text, the presence of graminoids and forbs had a similar effect on the $CO_2$ flux: their coverage correlated positively with both gross primary productivity and ecosystem respiration. This is why only the vascular leaf area index was needed to explain the $CO_2$ flux variation. For $CH_4$ flux, however, the relationship with LAI was found to be more complex, as $CH_4$ emissions increased with increasing graminoid LAI while the opposite was true for the forbs.

6. For future work it would be nice to have some isotopic data.

- We agree with this idea and will consider it for our future work.

7. Page 23 Line 636. It would be good here to have a paragraph or two that is more speculative (or perhaps in warning): we know that the climate in the Arctic and sub-Arctic is warming fast and changing – what is going to

happen?  Can this very detailed study give us any pointers to what is going to happen? The work in the paper is careful and well reported, but it needs to be given its wider context – Heiskanen et al are experts – what can they tell us about where these mires are going?

- To keep the conclusions clear and concise, we did not expand this section but added some text to Discussion. This additional discussion makes the connection between heatwaves, studied in the present manuscript, and Arctic warming more explicit; it also supports the final conclusion. We added the following paragraph to the end of Section 4.2.3:

"Heatwaves are predicted to become more frequent in the subarctic region as the climate warms (Masson-Delmotte et al., 2018). However, the impact of heatwaves on the C exchange of northern mires strongly depends on local soil moisture conditions. While drought leads to diminished C sequestration, warming accompanied by sufficient precipitation is likely to support the long-term peat accumulation in the subarctic, non-permafrost mires (Loisel et al., 2020). On the other hand, the vegetation composition and biomass production on these fens are susceptible to lowering water table level (Mäkiranta et al., 2018). Therefore, the functioning of subarctic fens may undergo substantial changes, if the water balance changes concurrently with the warming climate."

Masson-Delmotte, V., Zhai, P., Pörtner, H.-O., Roberts, D., Skea, J., Shukla, P. R., Pirani, A., Moufouma-Okia, W., Péan, C., Pidcock, R., Connors, S., Matthews, J. B. R., Chen, Y., Zhou, X., Gomis, M. I., Lonnoy, E., Maycock, T., Tignor, M. and Waterfield T. (Eds.): Global Warming of 1.5°C. An IPCC Special Report on the impacts of global warming of 1.5°C above pre-industrial levels and related global greenhouse gas emission pathways, in the context of strengthening the global response to the threat of climate change, sustainable development, and efforts to eradicate poverty, World Meteorological Organization, Geneva, Switzerland, 2018.

Loisel, J., Gallego-Sala, A. V., Amesbury, M. J. et al. Expert assessment of future vulnerability of the global peatland carbon sink. Nat. Clim. Change, doi:10.1038/s41558-020-00944-0, 2020.

Mäkiranta, P., Laiho, R., Mehtätalo, L., Straková, P., Sormunen, J., Minkkinen, K., Penttilä, T., Fritze, H. and Tuittila, E.: Responses of phenology and biomass production of boreal fens to climate warming under different water-table level regimes, Glob. Change Biol., 24, 944-956, doi:10.1111/gcb.13934, 2018.